

# Reeling them in: taxonomy of marine annelids used as bait by anglers in the Western Cape Province, South Africa

Carol Simon[1], Jyothi Kara[1,2], Alheit du Toit[1], Hendré van Rensburg[1], Caveshlin Naidoo[1] and Conrad A. Matthee[1]

[1] Department of Botany and Zoology, University of Stellenbosch, Stellenbosch, Western Cape, South Africa
[2] Research and Exhibitions, Iziko Museums of South Africa, Cape Town, Western Cape, South Africa

## ABSTRACT

**Background:** Common names are frequently used inconsistently for marine annelid species used as bait in the peer-reviewed literature, field guides and legislative material. The taxonomy of many such species based on morphology only also ignores cryptic divergences not yet detected. Such inconsistencies hamper effective management of marine annelids, especially as fishing for recreation and subsistence is increasing. This study investigates the scale of the problem by studying the use and names of bait marine annelids in the Western Cape Province of South Africa.

**Methods:** Fifteen recreational and six subsistence fishers at 12 popular fishing sites in the Western Cape Province donated 194 worms which they identified by common name. Worms were assigned scientific names according to a standard identification key for polychaetes from South Africa, and mitochondrial cytochrome oxidase I (COI) amplified and sequenced.

**Results:** This study identified 11 nominal species known by 10 common names, in the families Siphonosomatidae, Arenicolidae, Sabellaridae, Lumbrineridae, Eunicidae, Onuphidae and Nereididae. Cryptic diversity was investigated through employing mitochondrial COI sequences and these data will facilitate future identifications among widely distributed species. Several species (*Siphonosoma dayi*, *Abarenicola gilchristi*, *Scoletoma* species, *Marphysa corallina*, *Lysidice natalensis*, *Heptaceras quinquedens*, *Perinereis latipalpa*) are reported as bait for the first time, and while the names blood- and moonshineworms were consistently applied to members of Arenicolidae and Onuphidae, respectively, coralworm was applied to members of Sabellaridae and Nereididae. Analysis of COI sequences supported morphological investigations that revealed the presence of two taxonomic units each for specimens initially identified as *Gunnarea gaimardi* and *Scoletoma tetraura* according to identification keys. Similarly, sequences for *Scoletoma* species and *Lysidice natalensis* generated in this study do not match those from specimens in China and India, respectively. Further research is required to resolve the species complexes detected and also to refine the use of names by fishermen over a wider geographic range.

Corresponding author
Carol Simon, csimon@sun.ac.za

## INTRODUCTION

In South Africa, shore-based marine fishing is an important recreational activity and part of the livelihood for many subsistence fishermen and has shown a steady increase over the last decades (*McGrath et al., 1997*; *Sowman et al., 2014*; *Saayman et al., 2017*). There is a close link between shore fishing and bait collecting (*MacKenzie, 2005*), so an increase in fishing intensity will certainly correlate with an increase in harvesting of natural stocks of bait species (*Nel & Branch, 2014* cf. *Hodgson, Allanson & Cretchley, 2000*; *Napier, Turpie & Clark, 2009*; *Simon et al., 2019*). However, in a recent assessment of the impacts of recreational and subsistence fishing in marine ecosystems in South Africa, impacts of bait collecting received just a passing mention (*Majiedt et al., 2019*). This supports *Watson et al. (2017)*, who suggested that despite their wide use, marine annelids (*i.e.*, polychaete worms) and probably many other bait species are universally a poorly managed resource.

A wide variety of marine invertebrates are used as bait by South African fishermen (*MacKenzie, 2005*; *Branch et al., 2016*; *DAFF, 2017*). However, live marine annelids (indigenous or imported) are not sold in bait shops and are instead collected by subsistence and recreational fishermen who should possess appropriate permits (*DAFF, 2017*). Collection is controlled by taxon-specific daily limits (*DAFF, 2017*), but these restrictions have remained almost unchanged for decades (cf. *Gaigher, 1979*; *Van Herwerden et al., 1989* and *DAFF, 2017*). Furthermore, although nearly 2.5 million worms are harvested annually (*Turpie, Heydenrych & Lamberth, 2003*), biological information to inform management strategies is limited (*Gaigher, 1979*; *Van Herwerden, 1989*; *Lewis, 2005*; *Simon et al., 2020*) while restrictions on collection also do not accommodate the different bait collecting habits by recreational and subsistence fishermen (*Simon et al., 2019*). Knowing which species are being utilised is an important step towards improving management of a resource as many bait species, including those that may be morphologically very similar, may have different life history traits and habitat requirements (*Hutchings & Lavesque, 2021*), which may influence the vulnerability of species to exploitation.

The Marine Recreational Activity Information Brochure issued by the Department of Agriculture, Forestry and Fisheries in South Africa (now the Department of Forestry, Fisheries and Environment; *DAFF, 2017*) identifies bait worms generically as seaworms, polychaetes and flatworms, and by various common names. The only taxa identified by genus are *Arenicola Lamarck, 1801*, *Nereis Linnaeus, 1758*, *Pseudonereis Kinberg, 1865b* and *Gunnarea Johansson, 1927*. As no images are included in the brochure, it is unclear what the worms listed by common name are. However, the popular *Two Oceans: A guide to the Marine Life of southern Africa* (*Branch et al., 2016*) provides images and common and scientific names for some baitworms: bloodworm (*Arenicola loveni Kinberg, 1866*), musselworm (*Pseudonereis podocirra* (*Schmarda, 1861*) as *P. variegata* (Grube & Kröyer in *Grube, 1858*)), wonderworm (*Eunice aphroditois* (*Pallas, 1788*)), Cape reef worm

(*Gunnarea gaimardi* (*De Quatrefages, 1848*), as *G. capensis* (*Schmarda, 1861*) in earlier editions), and the estuarine wonderworm (*Marphysa haemasoma De Quatrefages, 1866*, as *M. elityeni Lewis & Karageorgopoulos, 2008*, see *Kara et al., 2020*). The species names for bloodworm, musselworm and Cape reef worm (also known as coralworm in *Branch et al., 2016*) correspond with those provided in the Government Gazette No. 39790 (*Marine Living Resources Act, 2014*). The latter source, however, uses different names for *E. aphroditois* (Bobbit or errant worm), *Arabella iricolor* (*Montagu, 1804*) (moonshineworm) and *M. haemasoma* (wonderworms and listed as *M. sanguinea* (*Montagu, 1813*)). The situation is further complicated by reports of bait worms in other sources; for example, *Diopatra Audouin & Milne Edwards, 1833* species have been called case worm (*Day, 1974*), moonshineworm (*Napier, Turpie & Clark, 2009*; *Van Rensburg, Matthee & Simon, 2020*), estuarine wonderworm (*Smith & Smith, 2012*; *Allanson, Human & Claassens, 2016*) and coralworm (*Fielding, 2007*; P.J. Fielding, 2017, personal communication), while *E. aphroditois* has also been called coralworm (*Wooldridge & Coetzee, 1998*). Thus, management of utilised worms may be hampered by confusion around the identities of the species that are harvested, and a lack of consensus in the names used among fishermen, scientists and managers active in South Africa.

The confusion around the use of common names is further complicated by recent taxonomic research which emphasised how poor our understanding of the biodiversity of South African marine annelids, including some used as bait, is. For example, *P. podocirra* and *M. haemasoma* were removed from synonymy with apparently globally widespread *P. variegata* and *M. sanguinea*, respectively, so both are in fact indigenous to South Africa (*Lewis & Karageorgopoulos, 2008*; *Kara, Macdonald & Simon, 2018*; *Kara et al., 2020*). By contrast, the *Diopatra* species used as bait in two estuaries on the south and southeast coasts of the country (*Van Der Westhuizen & Marais, 1977*; *Fielding, 2007*; *Napier, Turpie & Clark, 2009*; *Simon et al., 2019*), was only recently identified as *D. aciculata Knox & Cameron, 1971* (*Van Rensburg, Matthee & Simon, 2020*). This species was originally described in Australia (*Knox & Cameron, 1971*) and is probably alien in South Africa (*Elgetany et al., 2020*; *Van Rensburg, Matthee & Simon, 2020*). At least two other bait species, *A. iricolor* and *E. aphroditois*, are also apparently globally widespread with type localities geographically distant from South Africa (see *Day, 1967*), and may therefore either be misidentified indigenous, or unacknowledged alien species. Some species that are harvested (*e.g.*, *P. podocirra, E. aphroditois, A. iricolor, G. gaimardi*) are also widespread within South Africa (*Day, 1967*; *Branch et al., 2016*). The ranges of these species, which have planktonic larvae, span known phylogeographic barriers to gene flow in the region. It is thus likely that complexes of genetically distinct but morphologically identical or similar lineages exist (*i.e.*, cryptic or pseudocryptic species, respectively), each with discrete distributions. This was shown for species previously identified as *Pseudopolydora antennata* Claparède, 1869 from temperate and subtropical regions of the country (*Simon, Sato-Okoshi & Abe, 2019*), emphasising the need for thorough taxonomic studies of seemingly widespread species.

This study builds on taxonomic information gathered to date, and explores the use of common names and the nomenclature of marine annelid worms used as bait in the
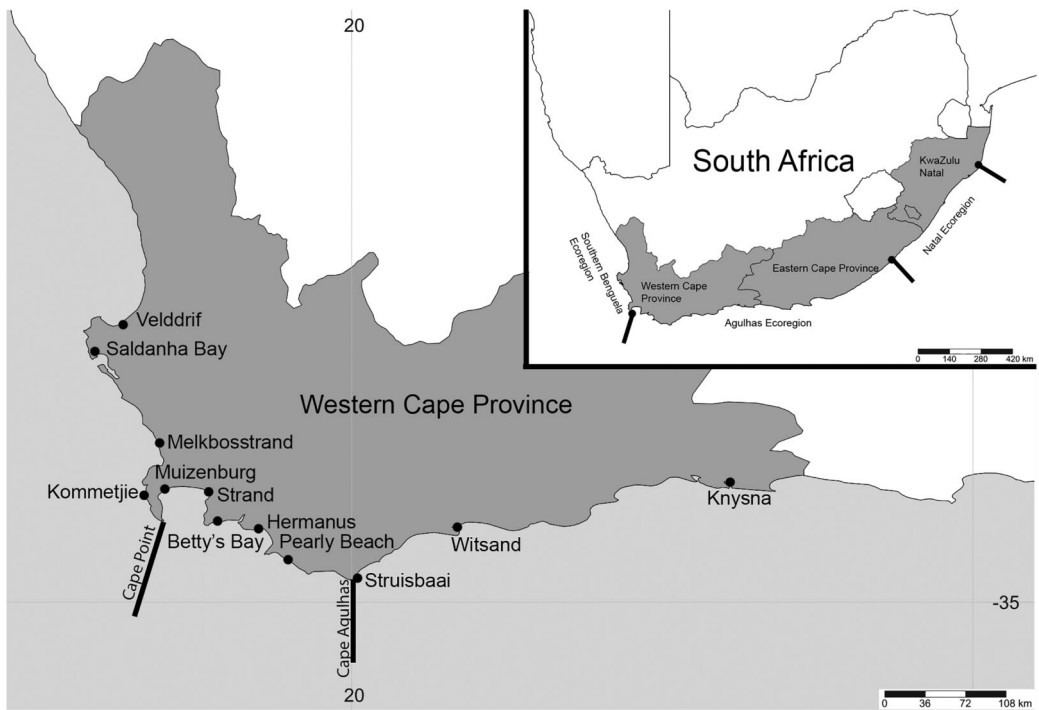

**Figure 1 Map of South Africa and the Western Cape province.** Sample sites in the Western Cape Province, South Africa: Velddrif, Saldanha Bay, Melkbosstrand, Kommetjie (Soetwater), Muizenberg, Strand, Betty's Bay, Hermanus, Pearly Beach, Struisbaai, Witsand, Knysna, with the two main barriers to gene flow in the Western Cape Province. Inset shows three main ecoregions along the South African coast. (Map created in http://www.simplemappr.net/ and edited in Photoshop Version C6.).

Western Cape Province where fishing is particularly popular (*Majiedt et al., 2019*), and where harvesting of worms is high (*Turpie, Heydenrych & Lamberth, 2003*). Furthermore, the province spans two vicariant barriers to gene flow at Cape Point and Cape Agulhas (*Teske et al., 2011*, Fig. 1), and this may also split species into different taxonomic units. The specific aims of the study are to: (1) identify and provide updated descriptions of the annelid species collected as bait by recreational and subsistence fishermen in the Western Cape Province of South Africa; (2) collate the common names used by the fishermen towards developing consensus for improved management; and (3) generate mtCOI sequences to explore the existence of species complexes locally and globally and facilitate identifications.

## METHODOLOGY

### Sample sites and collection

Sampling was conducted at 12 popular beach and estuarine fishing locations in the Western Cape Province, South Africa (Fig. 1), from June 2016 to May 2017. Collectively, these sites included sandy (Saldanha Bay, Muizenberg, Strand, Betty's Bay, Pearly Beach, Struisbaai, Witsand, Knysna) and or rocky (Velddrif, Melkbosstrand, Kommetjie, Betty's Bay, Hermanus, Witsand) habitats, which would influence the presence and absence of species collected. Worms were donated by bait collectors who all gave their

prior consent to participate in the project. Involvement by most recreational fishermen was confirmed prior to sampling *via* fishing mailing lists or word of mouth. Some additional recreational and all subsistence fishermen were approached on site. After the aims of the study were described to participants and verbal consent received (ethical clearance number: SU-HSD-001609 from Stellenbosch University), worms were collected according to the permitted methods (*DAFF, 2017*), under permit RES2017-27 issued to CS by the Department of Forestry, Fisheries and Environment. Additional samples of arenicolids (bloodworm) were collected by the authors using the same techniques (see *Simon et al., 2020*). The common names used by the bait collectors were noted for all worms. All sampling was conducted during low tide, with specific collection methods for the different taxa included in the systematic accounts below. In some instances, fishermen were only willing to donate a small piece of the worm that was sufficient for genetic analysis.

## Specimen identification and processing

Samples were relaxed in an isotonic solution of 7% $MgCl_2$ in tap water, measured and photographed. A section of each specimen from the mid-body or posterior was placed in 96% ethanol for molecular analysis. The rest of the specimen was fixed in 4% formalin in seawater for at least 2 days, washed in distilled water and stored in 70% ethanol. Samples were examined on Leica DM1000 light and MZ75 dissecting microscopes, and photographed with a Leica EC3 camera attachment, or on Leica DM750 light and M80 dissecting microscopes and photographed with an Olympus Targus TG5 attached to the microscope eyepieces. Where necessary, images were stacked in Helicon Focus Version 7.6.4 and processed in Photoshop Version C6. Specimens were identified using *Day (1967, 1974)*, and where necessary, more recent literature appropriate to individual taxa. All specimens were deposited at IZIKO South African Museum (Table 1).

## DNA extraction, amplification and sequencing

Approximately 25 mg of tissue was used either from mid-section or posterior end to extract DNA using the Zymo Quick DNA™ MiniPrep Plus kit (Zymo-Spin™) and according to the manufacturer's protocol. The universal primer pair: LCO1490 and HCO2198 (*Folmer et al., 1994*) was used to amplify a fragment of the cytochrome oxidase subunit 1 (COI) gene for all species. The following PCR thermal conditions were used: 94 °C for 3 min; 34 cycles with 94 °C for 45 s, 42 °C for 1 min and 72 °C for 1 min and a final extension at 72 °C for 7 min (*Bleidorn, Vogt & Bartolomaeus, 2005*). The amplified PCR products were visualised on a 1% agarose gel using 3 µl of PCR product and 5 µl of Quick-Load Purple 100 bp DNA ladder (New England BioLabs Inc.), following *Simon et al. (2020)*. All PCR products were sequenced using Sanger sequencing at the Central Analytical Facility at Stellenbosch University. All newly generated sequences were uploaded on GenBank (Table 1).

**Table 1 Museum and GenBank accession numbers with location and collector details.**

| Species name | Common name according to fisherman[$] | Location | Fisherman's name | Type of Bait collector | Collector and sample processor | GenBank accession number (COI) (number of individuals) | Museum accession number (number of individuals) |
|---|---|---|---|---|---|---|---|
| *Siphonosoma dayi* | Sandworm | Knysna Estuary | Gerrie Barnard | Recreational | AdT | MW598440 | MB-A090313 |
| | Sandworm | Knysna Estuary | Gerrie Barnard | Recreational | AdT | MW598441 | MB-A090318 |
| *Abarenicola gilchristi* | Bloodworm | Betty's Bay | Ethan Newman | Recreational | EN & CS | NS | MB-A090223 - MB-A090226 (4) |
| | Bakkiewurm | Pearly Beach | Frans | Recreational | CN & AdT | MW595992 | MB-A090249 |
| | Bakkiewurm | Pearly Beach | Frans | Recreational | CN & AdT | MW595993 | DNA only |
| | Bakkiewurm | Pearly Beach | Frans | Recreational | CN & AdT | MW595994 | DNA only |
| | Bakkiewurm | Pearly Beach | Frans | Recreational | CN & AdT | MW595995 | DNA only |
| *Arenicola loveni*[#] | Bloodworm | Betty's Bay | Morne & Victor | Recreational | AdT | MK922184 | MB-A090220 |
| | Bloodworm | Betty's Bay | Morne & Victor | Recreational | AdT | MK922185 | MB-A090221 |
| | Bloodworm | Betty's Bay | Morne & Victor | Recreational | AdT | MK922163 | MB-A090222 |
| | Blood worm | Knysna Estuary | Gerrie Barnard | Recreational | AdT | MK922157 | MB-A090231 |
| | Blood worm | Knysna Estuary | Gerrie Barnard | Recreational | AdT | MK922158 | MB-A090232 |
| | Blood worm | Knysna Estuary | Gerrie Barnard | Recreational | AdT | MK922159 | MB-A090233 |
| | Blood worm | Knysna Estuary | Dewald Kamp | Recreational | AdT | MK922160 | MB-A090234 |
| | Blood worm | Knysna Estuary | Dewald Kamp | Recreational | AdT | MK922161 | MB-A090235 |
| | Blood worm | Knysna Estuary | Albert Kapp | Recreational | AdT | MK922158 | MB-A090236, MB-A090237 |
| | Bloodworm | Muizenberg | Anonymous | Recreational | AdT & CN | MK922158 | MB-A090227, MB-A090229 |
| | Bloodworm | Muizenberg | Anonymous | Recreational | AdT & CN | MK922164 | MB-A090228 |
| | Bloodworm | Muizenberg | Anonymous | Recreational | AdT | NS | MB-A090230 |
| | Bloodworm | Muizenberg | Anonymous | Recreational | AdT & CN | NS | MB-A090374 |
| | Bloodworm | Pearly Beach | Ferdi Joubert | Recreational | AdT & HvR | MK922163 | MB-A090246, MB-A090247 |
| | Bloodworm | Pearly Beach | Ferdi Joubert | Recreational | AdT & HvR | MK922183 | MB-A090248 |
| | Bloodworm | Saldanha Bay | Anonymous | Unspecified | CN | MK922165 | MB-A090257 |
| | Bloodworm | Saldanha Bay | Anonymous | Unspecified | CN | MK922166 | MB-A090258, MB-A090264 |
| | Bloodworm | Saldanha Bay | Anonymous | Unspecified | CN | MK922167 | MB-A090259 |
| | Bloodworm | Saldanha Bay | Anonymous | Unspecified | CN | MK922168 | MB-A090260 |
| | Bloodworm | Saldanha Bay | Anonymous | Unspecified | CN | MK922169 | MB-A090261 |
| | Bloodworm | Saldanha Bay | Anonymous | Unspecified | CN | MK922170 | MB-A090262 |
| | Bloodworm | Saldanha Bay | Anonymous | Unspecified | CN | MK922171 | MB-A090263, MB-A090266 |
| | Bloodworm | Saldanha Bay | Anonymous | Unspecified | CN | MK922172 | MB-A090265 |
| | Bloodworm | Saldanha Bay | Anonymous | Unspecified | CN | NS | MB-A090375 |
| | Bloodworm | Struisbaai | Gert Kotze | Recreational | CN, AdT & HvR | MK922163 | MB-A090238, MB-A090242 |
| | Bloodworm | Struisbaai | Gert Kotze | Recreational | CN & AdT | MK922173 | MB-A090239 |

| Species name | Common name according to fisherman[$] | Location | Fisherman's name | Type of Bait collector | Collector and sample processor | GenBank accession number (COI) (number of individuals) | Museum accession number (number of individuals) |
|---|---|---|---|---|---|---|---|
| | Bloodworm | Struisbaai | Gert Kotze | Recreational | CN & AdT | MK922174 | MB-A090240 |
| | Bloodworm | Struisbaai | Gert Kotze | Recreational | CN & AdT | MK922158 | MB-A090241 |
| | Bloodworm | Struisbaai | Gert Kotze | Recreational | CN | MK922175 | MB-A090243 |
| | Bloodworm | Struisbaai | Gert Kotze | Recreational | CN | MK922176 | MB-A090244 |
| | Bloodworm | Struisbaai | Gert Kotze | Recreational | CN | MK922158 | MB-A090245, MB-A090250, MB-A090251, MB-A090254, MB-A090255 |
| | Bloodworm | Witsand | Paul | Recreational | CN | MK922178 | MB-A090252 |
| | Bloodworm | Witsand | Paul | Recreational | CN | MK922179 | MB-A090253 |
| | Bloodworm | Witsand | Paul | Recreational | CN | MK922157 | MB-A090256 |
| *Gunnarea gaimardi* | Coralworm | Betty's Bay | Morne & Victor | Recreational | AdT | MN045177 | DNA only |
| | Coralworm | Betty's Bay | Morne & Victor | Recreational | AdT | MN045178 | DNA only |
| | Coralworm | Betty's Bay | Morne & Victor | Recreational | AdT | MN045179 | DNA only |
| | Coralworm | Betty's Bay | Ethan Newman | Recreational | CS | MN045177 | MB-A090336, MB-A090337, MB-A090339 |
| | Coralworm | Betty's Bay | Ethan Newman | Recreational | CS | MN045181 | MB-A090340 |
| | Coralworm | Betty's Bay | Ethan Newman | Recreational | CS | MN045180 | MB-A090441 |
| | Polwurm | Hermanus | Hein Engelbrecht | Recreational | AdT & HvR | MN045177 | MB-A090341, MB-A090342, MB-A090344, MB-A090345, MB-A090347, MB-A090348 |
| | Polwurm | Hermanus | Hein Engelbrecht | Recreational | AdT & HvR | NS | MB-A090343 |
| | Polwurm | Hermanus | Hein Engelbrecht | Recreational | AdT & HvR | MN045182 | MB-A090346 |
| | Coralworm | Velddrif | Anonymous | Subsistence | AdT | MN045177 | MB-A090356 - MB-A090358, MB-A090364, MB-A090367 - MB-A090371 (9) |
| | Coralworm | Velddrif | Anonymous | Subsistence | AdT | MN045179 | MB-A090360 |
| *Gunnarea* sp.1 | Coralworm | Witsand | Paul | Recreational | AdT | MN045184 | MB-A090293 |
| | Coralworm | Witsand | Paul | Recreational | AdT | MN045183 | MB-A090294 |
| *Scoletoma* sp. 1 (Betty's Bay) | Puddingworm | Betty's Bay | Ethan Newman | Recreational | CS | MN419154 | MB-A090332 |
| *Scoletoma* sp. 2 (Hermanus) | Puddingworm | Hermanus | Hein Engelbrecht | Recreational | AdT & HvR | NS | MB-A090349 |
| | Puddingworm | Hermanus | Hein Engelbrecht | Recreational | AdT & HvR | MN419157 | MB-A090350 |

*(Continued)*

| Species name | Common name according to fisherman[$] | Location | Fisherman's name | Type of Bait collector | Collector and sample processor | GenBank accession number (COI) (number of individuals) | Museum accession number (number of individuals) |
|---|---|---|---|---|---|---|---|
| | Puddingworm | Hermanus | Hein Engelbrecht | Recreational | AdT & HvR | NS | MB-A090351 |
| | Puddingworm | Hermanus | Hein Engelbrecht | Recreational | AdT & HvR | NS | MB-A090352 |
| | Puddingworm | Hermanus | Hein Engelbrecht | Recreational | AdT & HvR | MN419156 | MB-A090353 |
| | Puddingworm | Hermanus | Hein Engelbrecht | Recreational | AdT & HvR | MN419155 | MB-A090354 |
| *Marphysa cf. corallina* | Wonderworm | Witsand | Paul | Recreational | AdT | MN067881 | MB-A090276 - MB-A090278, MB-A090280 (4) |
| | Wonderworm | Witsand | Paul | Recreational | AdT | MN067882 | MB-A090279 |
| *Marphysa haemasoma* | Wonderworm | Betty's Bay | Ethan Newman | Recreational | CS | NS | MB-A090331 |
| | Wonderworm | Betty's Bay | Ethan Newman | Recreational | CS | MN067877 | MB-A090333, MB-A090335, MB-A090338 (3) |
| | Wonderworm | Betty's Bay | Ethan Newman | Recreational | CS | NS | MB-A090334 |
| | Wonderworm | Knysna Estuary | Anonymous | Recreational | AdT | MN067879 (3) | DNA only |
| | Wonderworm | Knysna Estuary | Anonymous | Recreational | AdT | MN067878 (2) | DNA only |
| | Bloukoppie | Knysna Estuary | Anonymous | Subsistence | AdT | MN067878 | MB-A090326, MB-A090328 (2) |
| | Bloodworm | Melkbos Strand | Lucas | Subsistence | AdT & CN | MN067877 (2) | DNA only |
| | Bloodworm | Melkbos Strand | Lucas | Subsistence | AdT & CN | MN067877 | MB-A090267 - MB-A090270 (4) |
| | Wonderworm | Soetwater Kommetjie | Altus | Subsistence | AdT | MN067877 | DNA only |
| | Wonderworm | Soetwater Kommetjie | Altus | Subsistence | AdT | NS | MB-A090272 |
| | Wonderworm | Soetwater Kommetjie | Altus | Subsistence | AdT | MN067877 | MB-A090273 - MB-A090275, MB-A090317 (4) |
| | Wonderworm | Strand | Marnus | Subsistence | AdT & HvR | MN067880 | DNA only |
| | Wonderworm | Strand | Marnus | Subsistence | AdT & HvR | MN067880 | MB-A090271, MB-A090315 (2) |
| *Lysidice natalensis* | Musselworm | Witsand | Paul | Recreational | AdT | MN419162 | MB-A090281 |
| | Musselworm | Witsand | Paul | Recreational | AdT | MN419168 | MB-A090282 |
| | Musselworm | Witsand | Paul | Recreational | AdT | MN419165 | MB-A090283, MB-A090285 (2) |
| | Musselworm | Witsand | Paul | Recreational | AdT | MN419164 | MB-A090284 |
| | Musselworm | Witsand | Paul | Recreational | AdT | MN419160 | MB-A090286 |
| | Musselworm | Witsand | Paul | Recreational | AdT | MN419161 | MB-A090287 |
| | Musselworm | Witsand | Paul | Recreational | AdT | MN419158 | MB-A090288 |
| | Musselworm | Witsand | Paul | Recreational | AdT | MN419159 | MB-A090289 |
| | Musselworm | Witsand | Paul | Recreational | AdT | MN419167 | MB-A090291 |
| | Musselworm | Witsand | Paul | Recreational | AdT | MN419163 | MB-A090292 |

| Species name | Common name according to fisherman[$] | Location | Fisherman's name | Type of Bait collector | Collector and sample processor | GenBank accession number (COI) (number of individuals) | Museum accession number (number of individuals) |
|---|---|---|---|---|---|---|---|
| *Heptaceras quinuedens* | Moonshineworm | Pearly Beach | Ferdi Joubert | Recreational | AdT & HvR | NS | MB-A090432 - MB-A090436 (5) |
| | Moonshineworm | Strand | Hermann Schuch & Charlie Friess | Recreational | AdT & HvR | NS | MB-A090442 |
| | Moonshineworm | Struisbaai | Gert Kotze | Recreational | CN, AdT & HvR | NS | MB-A090421 - MB-A090431, MB-A090437 - MB-A090440 (15) |
| *Perinereis latipalpa* | Coralworm | Kommetjie | Mario | Subsistence | AdT | NS | MB-A090297 - MB-A090299 (3) |
| *Pseudonereis podocirra* | Musselworm | Betty's Bay | Morne & Victor | Recreational | AdT | MN067871 | MB-A090302, MB-A090305 (2) |
| | Musselworm | Betty's Bay | Morne & Victor | Recreational | AdT | MN067870 | MB-A090304 |
| | Musselworm | Hermanus | Hein Engelbrecht | Recreational | AdT & HvR | MN067872 | MB-A090306 |
| | Musselworm | Hermanus | Hein Engelbrecht | Recreational | AdT & HvR | MN067873 | MB-A090307 |
| | Musselworm | Hermanus | Hein Engelbrecht | Recreational | AdT & HvR | MN067871 | MB-A090308, MB-A090309, MB-A090443 |
| | Musselworm | Hermanus | Hein Engelbrecht | Recreational | AdT & HvR | MN067872 | MB-A090310 |
| | Coralworm | Velddrif | Anonymous | Subsistence | AdT | MN067874 | MB-A090355, MB-A090362, MB-A090363, MB-A090365 (4) |
| | Coralworm | Velddrif | Anonymous | Subsistence | AdT | MN067871 | MB-A090359, MB-A090361 (2) |
| | Coralworm | Velddrif | Anonymous | Subsistence | AdT | MN067872 | MB-A090366 |
| | Coralworm | Velddrif | Anonymous | Subsistence | AdT | MN067875 | MB-A090372 |
| | Coralworm | Velddrif | Anonymous | Subsistence | AdT | MN067876 | MB-A090373 |

**Notes:**
[#] Sequences were previously published in *Simon et al. (2020)*.
[$] The English names are listed, although fishermen frequently use Afrikaans translations: bloodworm (bloedwurm), Coral worm (koraalwurm), mussel worm (mosselwurm), moonshine worm (maanskynwurm), pudding worm (poedingwurm), wonderworm (wonderwurm). English names were never used for polwurm or bakkiewurm.
Baitworm species from Western Cape, South Africa, found in this study, including common names, locations, collector details. GenBank accession numbers may be repeated when haplotypes are shared among different individuals. Samples were received from contributing fishermen and processed by Alheit du Toit (AdT), Caveshlin Naidoo (CN), Carol Simon (CS), Ethan Newman (EN) and Hendré van Rensburg (HvR). NS, no sequences.

## Molecular analysis

Sequences were edited in BioEdit Version 7.2.6 (*Hall, 1999*) and aligned using ClustalW with default parameters in MEGA X (*Kumar et al., 2018*). Neighbour joining trees were constructed in the same program, per family. Nodal support was obtained using 10 000 bootstrap replicates using the maximum composite likelihood method, with uniform rates and pairwise deletion.

## RESULTS

Worms were donated by 15 recreational and six subsistence fishers, with two additional fishers who were not categorised (Table 1). In total, these fishers donated 194 specimens belonging to seven families and 11 nominal species: *Siphonosoma dayi Stephen, 1942*, *Abarenicola gilchristi Wells, 1963*, *Arenicola loveni*, *Gunnarea gaimardi*, *Scoletoma* cf. *tetraura*, *Marphysa* cf. *corallina*, *M. haemasoma*, *Lysidice natalensis* Kinberg, 1865, *Heptaceras quinquedens* (*Day, 1951*), *Perinereis latipalpa* (*Schmarda, 1861*) and *Pseudonereis podocirra* (Table 1). Together, these species were referred to by 10 common names (Table 1). Sequences could not be generated for *Heptaceras quinquedens* and *Perinereis latipala* even after multiple attempts, with the remaining nine species representing 11 genetically distinct species, including two species each of *Gunnarea* and *Scoletoma* (Table 1).

Taxonomic account
Order: Sipuncula *Stephen, 1964*
Family: Siphonosomatidae *Kawauchi, Sharma & Giribet, 2012*
Genus: *Siphonosoma Spengel, 1912*
Species: *Siphonosoma dayi Stephen, 1942*
Figures 2 & 3

*Siphonosoma dayi Stephen, 1942*: 246–247, Pl. XI, Figs. 1 & 2; *Day, 1974*: 49
Common name: Sandworm.

*Material examined*: Knysna: 34°03′56.0″S 23°02′57.4″E, 2 specimens, MB-A090313 and MB-A090318, 27 January 2017, coll. A. du Toit, mid-intertidal sandflats in estuary.

*Description*: Trunk length 198 and 230 mm, introvert of former 17 mm. In life body light to dark pink, colour retained after fixation (Fig. 2A), internally pearlescent pink (Figs. 2E–2H). Skin covered with oval shaped papillae in longitudinal rows, following contours of circular muscle, appear white after fixation. Introvert has terminal mouth ringed with short tentacles (Fig. 2B); papillae chitinised, tubular, scale-like and with dark edges arranged in rows on circular muscle bands (Figs. 2B–2D); larger and more numerous in anterior end (Fig. 2C) than posterior (Fig. 2D). Longitudinal muscle-layer divided into 21 or 22 bands (Figs. 2E, 2G, black arrows), anastomosing anteriorly to form single sheet in region of introvert (Fig. 2E, black arrowhead). Four retractor muscles; dorsal pair attached to body wall anteriorly, ventral pair attached more posteriorly (Figs. 2E, 2G, 2H white arrowheads). Two branches of spindle muscle insert close to dorsal retractor muscles (Fig. 2H, black arrowhead). One pair of nephridia (Figs. 2E, 2F).

*Remarks*: New specimens match the original description by *Stephen (1942)*. Although only two specimens were collected and sequenced, *S. dayi* (Fig. 3) forms a well-supported clade which is independent from other known species within the genus.

*Collection method*: Hand digging and pumping.

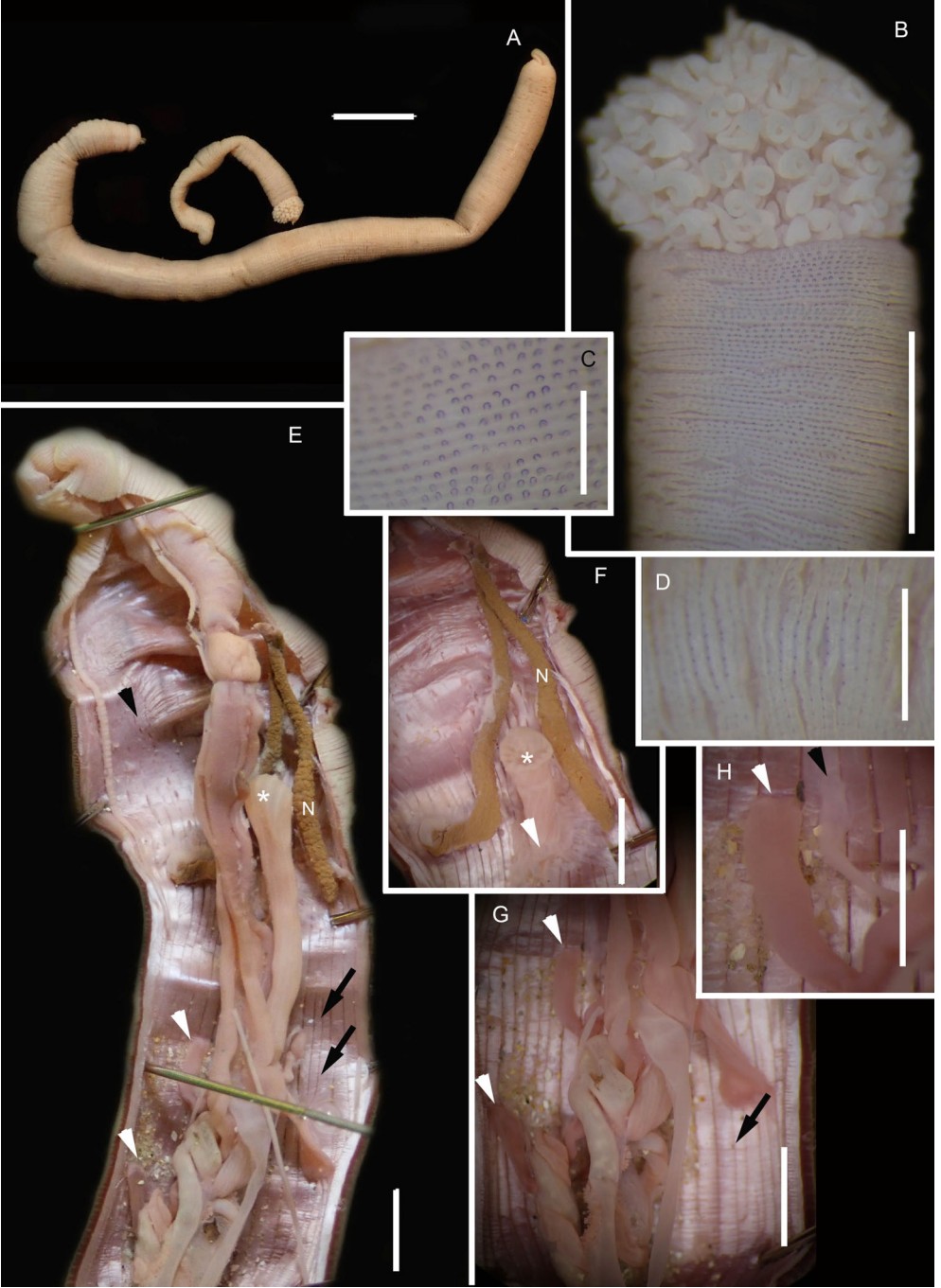

**Figure 2 Morphology of *Siphonosoma dayi* Stephen, 1942.** (A) Fixed specimen, in two pieces, (B) Everted introvert with tentacles, (C) Scales on anterior of introvert, (D) scales on posterior of introvert, (E) Anterior, internal structure showing insertions of introvert muscles (white arrowheads), bands of longitudinal muscles (black arrows), anastomosed sheet of muscle in anterior (black arrowhead) and rectum (*), (F) Pair of nephridia (N) and broken rectum (*) with insertion of anus (white arrowhead), (G) Magnification of digestive system showing insertions of introvert muscles (arrowheads) and bands of longitudinal muscle (black arrow), (H) Close-up of insertion of dorsal introvert muscle (white arrowhead) and spindle muscle (black arrowhead). Scale Bars: (A) = 10 mm; (B), (H) = 2.5 mm; (C) = 0.5 mm, (D) = 1 mm; (E), (F), (G) = 5 mm; (A), (B)–(D) = MB A090318; (E)–(H) = MB A090313.

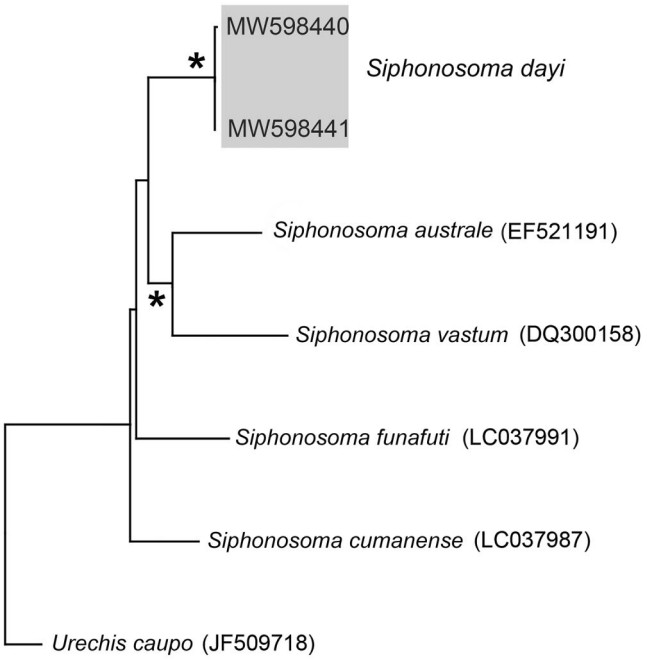

**Figure 3 Neighbour Joining tree using mitochondrial sequences of various *Siphonosoma Spengel, 1912* species, including *S. dayi Stephen, 1942* from Knysna.** *Indicates bootstrap support greater than or equal to 80%. Sequences obtained in this study are highlighted in grey. *Urechis caupo Fisher & MacGinitie, 1928* was used as the outgroup. Scale bar represents number of substitutions.

*Type locality*: Knysna, Western Cape Province, South Africa.
*Known distribution in South Africa*: Knysna (*Day, 1974*).
*Ecology*: In sand in low to mid intertidal in estuary.

Subclass: Sedentaria *Lamarck, 1818*
Infraclass: Scolecida *Rouse & Fauchald, 1997*
Family: Arenicolidae *Johnston, 1835*
Genus: *Arenicola Lamarck, 1801*
Species: *Arenicola loveni Kinberg, 1866*
Figures 4–6A & 6B

*Arenicola loveni Kinberg, 1866*: 355; *Ashworth, 1911*: 2–17, Figs. 1–3; *Wells 1962*: 348, Pl. 2 & 4; *Day, 1967*: 610, Fig. 29.1 f–k; *Day, 1974*: 62, Fig. 54; *Branch et al., 2016*: 72, Fig. 27.9
Common name: Bloodworm.

*Material examined*: Betty's Bay: 34°22′39.6″S 18°51′21.6″E, 3 specimens, MB-A090220–MB-A090222, 10 February 2017, mid-intertidal, sandy beach, coll. A. du Toit. Knysna: 34°03′

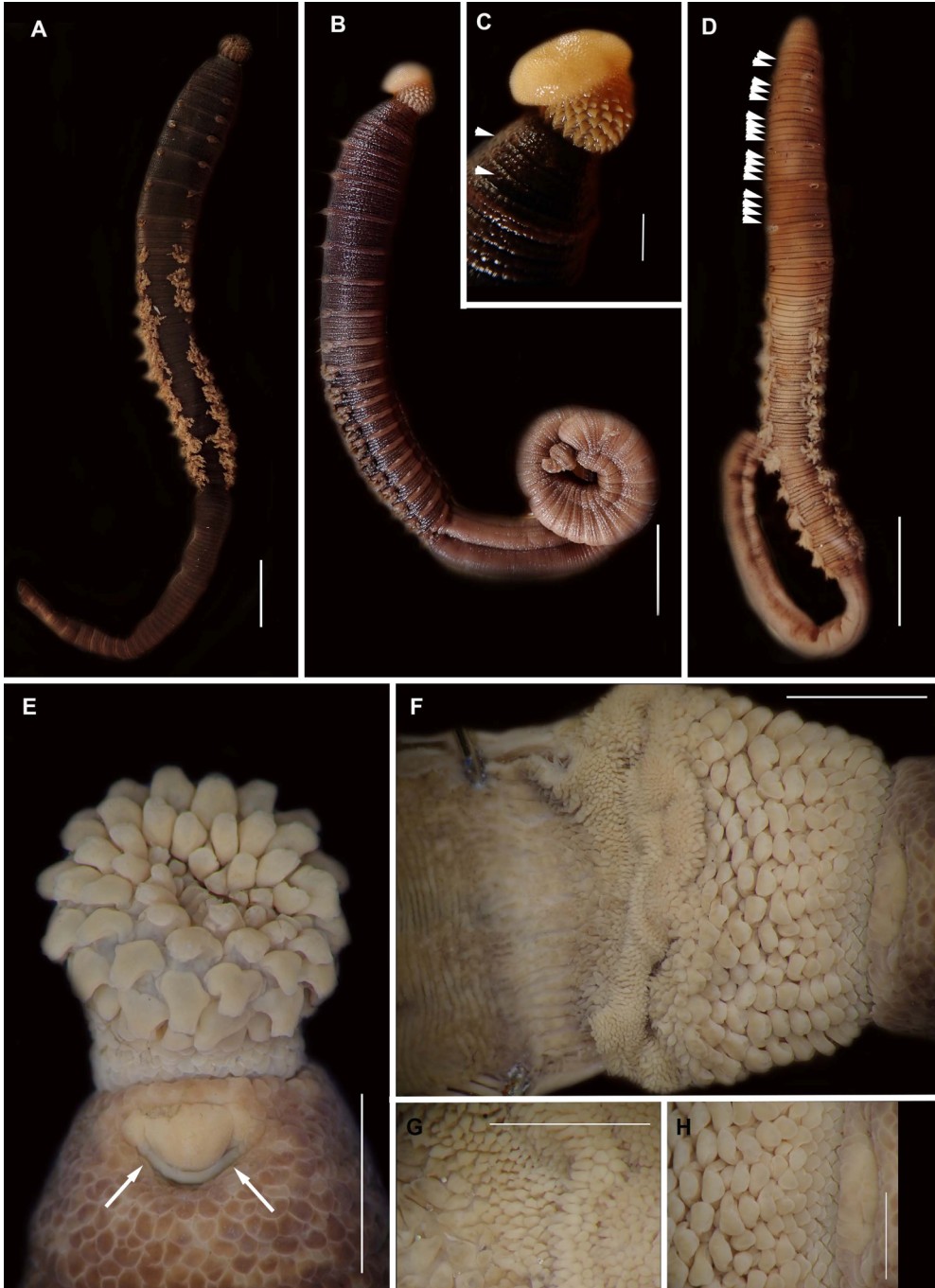

**Figure 4 External morphology of *Arenicola loveni Kinberg, 1866*.** *(A)* Almost uniformly dark specimen from Muizenberg, dorsal view, (B) Dark specimen with distinctly lighter tail from Struisbaai, lateral view, (C) Close-up of proboscis of specimen in (B) showing annuli (white arrowheads), (D) Light brown specimen with distinctly lighter branchial and tail region from Muizenberg, showing annuli in anterior chaetigers (white arrowheads), dorsal view, (E) Prostomium and partially everted proboscis, dorsal view, arrows show nuchal grooves, (F) Proboscis showing papillae in different regions, dorsal view, (G) Papillae of distal part of proboscis, dorsal view, (H) Papillae of proximal part of proboscis, dorsal view. Scale bars: (A), (B), (D) = 2 cm, (C), (E), (F) = 5 mm, (G), (H) = 2.5 mm. (A) = MB-A090229, (B) = MB-A090241, (D) = MB-A090227, (E)–(H): MB-A090259     

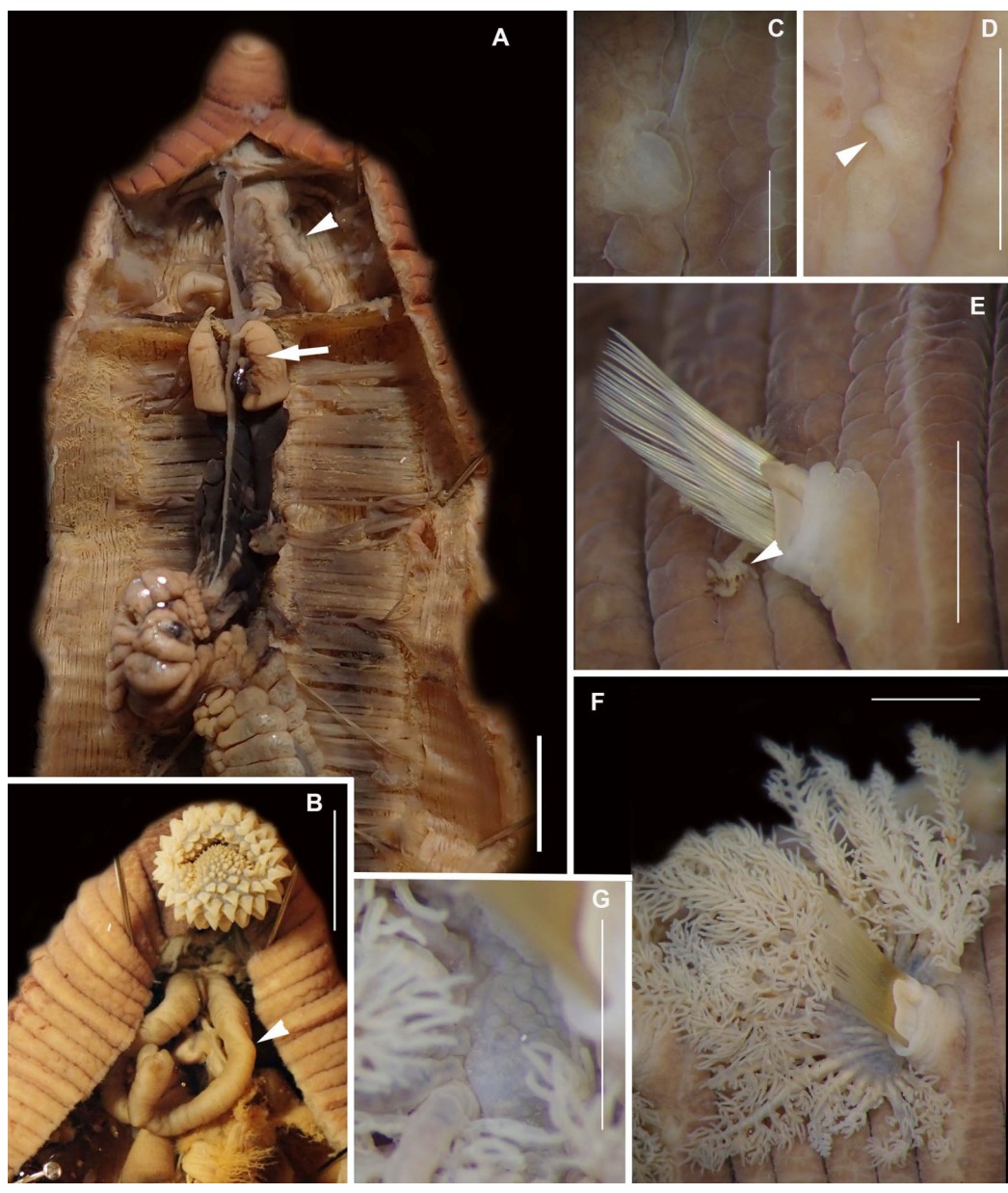

**Figure 5 Morphology of *Arenicola loveni* Kinberg, 1866.** *(A)* Dorsal view of digestive system with septal pouches (arrowhead) and single pair of oesophageal caecae (arrow), (B) Close up of anterior digestive system (ventral view) and septal pouches (arrowhead) and partially everted proboscis, (C) Hooded nephridiopore, (D) Partially hooded nephridiopore, (E) Chaetiger 7 with vestigial branchia (arrowhead), (F) Fully formed branchiae on chaetiger 14, (G) Close up of palmar membrane showing papillated surface. Scale bars: (A), (B), = 10 mm, (C), G = 1.5 mm, (D)–(F) = 2.5 mm, (A): MB-A090252, (B), (D): MB-A090250, (C), (E)–(G): MB-A090259.

28.6″S 23°02′30.9″E, 3 specimens, MB-A090231–MB-A090233, 27 January 2017, 34°03′54.3″ S 23°03′03.7″E, 2 specimens MB-A090234–MB-A090235, 28 January 2017, 2 specimens, 34°03′54.3″S 23°03′03.7″E, 29 January 2017, MB-A090236–MB-A090237, mid-intertidal sandy beach, coll. A. du Toit. Muizenberg, 34°06′18.7″S 18°28′47.4″E, 1 specimen, MB-A090230, 13 March 2017, coll. A. du Toit, 34°06′27.6″S 18°28′22.3″E, 1 specimen,

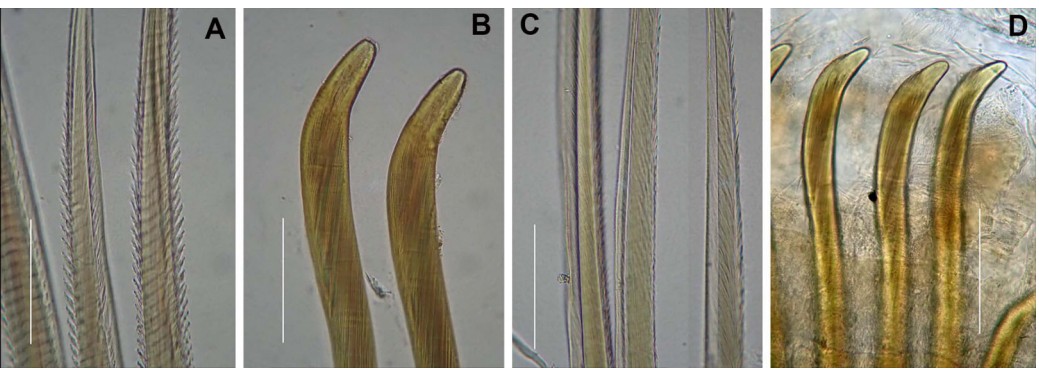

**Figure 6 Arenicolid chaetae.** (A) Notochaetae and (B) Neuropodial hooks of *Arenicola loveni* Kinberg, 1866, (C) Notochaetae and (D) Neuropodial hooks of *Abarenicola gilchristi* Wells, 1963. Scale Bars: (A)–(D) = 0.1 mm. (A), (B) = MB-A090261, (C), (D) = MB-A090225.

MB-A090227, 2 specimens, 34°06′18.7″S 18°28′47.4″E, MB-A090228–MB-A090229, 25 February 2017, 34°06′27.6″S 18°28′22.3″E, 1 specimen, MB-A090374, 25 February 2017, coll. A. du Toit and C. Naidoo; low intertidal in surf zone, sandy beach. Pearly Beach: 34°39′33″S 19°29′27.43.6″E, 3 specimens MB-A090246–MB-A090248, 12 February 2017, coll. A. du Toit and H. van Rensburg, low-intertidal, sandy beach. Saldanha Bay: 33°00′26.9″S 17°56′46.3″E, 7 specimens, MB-A090257–MB-A090263, 27 May 2017; 32°59′49.3″S 17°57′58.3″E, 3 specimens, MB-A090264–MB-A090266, 27 May 2017, 33°00′26.9″S 17°56′46.3″E, 1 specimen, MB-A090375, 27 May 2017, coll. C. Naidoo, low intertidal, sandy beach. Struisbaai: 34°47′41.1″S 20°02′57.6″E, 1 specimen, MB-A090238, 12 February 2017; 3 specimens, MB-A090239–MB-A090241, 10 April 2017, 4 specimens, MB-A090242–MB-A090245, coll. C. Naidoo, A. du Toit and H. van Rensburg, mid to low intertidal, sandy beach. Witsand: 34°23′59.9″S 20°49′47.5″E, 7 specimens, MB-A090250–MB-A090256, coll. C. Naidoo, low intertidal, sandy beach.

*Description*: Live specimens up to 580 mm, including tail. Fixed specimens up to 296 mm long (excluding achaetous tail), 19.2 mm wide at chaetiger 1. In life, body colour variable; pink to brown, dark brown to black; usually darker in anterior, becoming lighter from branchial region posteriorly (Figs. 4A, 4B, 4D), colour retained when fixed. Epidermis tessellated to chaetiger 5 or 6, papillated from chaetae 6 or 7 onward, including achaetous tail. Chaetigerous annuli prominent, number of annuli between first 4 chaetigers 2-3-4, thereafter 4 (Fig. 4D).

Anterior region consists of trilobed, non-retractable prostomium with nuchal groove on each side (Fig. 4E, arrows). One achaetous segment with 2 annuli (Fig. 4C). Proboscis eversible; covered with papillae, no pigment (Figs. 4C, 4E–4H). Papillae on proximal section large and triangular (Figs. 4C, 4E, 4F, 4H). Papillae in median section more densely packed, small and nipple-shaped, becoming larger and more conical distally (Fig. 4C, 4F, 4G). One pair of long septal pouches that reach back to at least third diaphragm (Figs. 5A, 5B). One pair of conical oesophageal caecae (Fig. 5A). Thorax with 19 chaetigers. Notopodia rounded triangles, retractable lobes in oval torus (Fig. 5E). Notochaetae

capillaries in two rows, anterior row shorter than posterior; with lateral toothed-crests and spinulose lamina (Fig. 6A). Neuropodia oval bearing single row of unidentate hooks (Fig. 6B), sometimes with faint denticle. Neuropodia long, approach midline of venter in branchiate region. Branchiae on chaetigers 7–19 (13 pairs), highly vascularised, highly branched, arborescent (tree-shaped) (Fig. 5F). On chaetiger 7 branchiae vestigial; 2–10 short gill stems, palmar membrane sometimes inconspicuous (Fig. 5E). Up to 22 main gill stems on branchiae on chaetigers 8 to 18, usually fewer on chaetiger 19. Palmar membrane fuse lower third of gill stems (Fig. 5F), sometimes papillated (Fig. 5G). Five pairs of nephridia on chaetigers 5–9; nephridiopores hooded, partially hooded (Figs. 5C, 5D) or unhooded, posterior to dorsal end of neuropodium. Tail achaetous, papillated, anus terminal.

*Remarks*: Specimens examined here conform to descriptions by *Ashworth (1911)* and *Wells (1962)* which included type material, but maximum size is larger. However, oval depressions seen by *Ashworth (1911)* ventral to some notopodia were not observed. The colour variants of *A. loveni* from all sites form a well-supported clade (Fig. 7) which is exemplified by the fact that those illustrated in Figs. 4A, 4B and 4D are represented by an identical sequence (MK 922158). This clade includes two subclades, representing specimens collected on the west and south coasts, respectively. The structure seen here was previously reported in *Simon et al. (2020)*, where nuclear data confirmed that these west and south coast clades represent a single species. The separation between these clades is demarcated by Cape Point, a location known to present a barrier to gene flow (*Teske et al., 2011*; *Simon et al., 2020*).

*Type locality*: Durban, KwaZulu-Natal, South Africa.
*Collection method*: By pump or digging with hand or trowel and hooking out with a wire. In Muizenburg collected from within surfzone.
*Known distribution in South Africa*: Saldanha Bay (Western Cape Province) to Durban (Kwa-Zulu Natal) (*Day, 1967*).
*Ecology*: In sand in low to mid intertidal on sheltered sandy shores and estuaries.

Genus: *Abarenicola Wells, 1959*
Species: *Abarenicola gilchristi Wells, 1963*
Figures 6C & 6D & 8

*Abarenicola gilchristi Wells, 1963*: 147–149, Fig. 6c, Pl. 2 & 5; *Day, 1967*: 611–612, Fig. 29.2
*Arenicola assimilis* var. *affinis Ashworth, 1911*: 18, Figs 4 & 5 (in part); *Day 1955*: 427
Common name: Bloodworm, bakkiewurm.

*Material examined*: Betty's Bay: 34°22′S 18°51′E, 4 specimens (incomplete) (MB-A090223–MB-A090226), 3 June 2016, mid-intertidal, sand, coll. E. Newman. Pearly Beach: 34°39′48.4″ S 19°29′17.2″E, 1 specimen (MB-A090249), 10 April 2017, low-intertidal, sand, coll. A. du Toit and C. Naidoo.

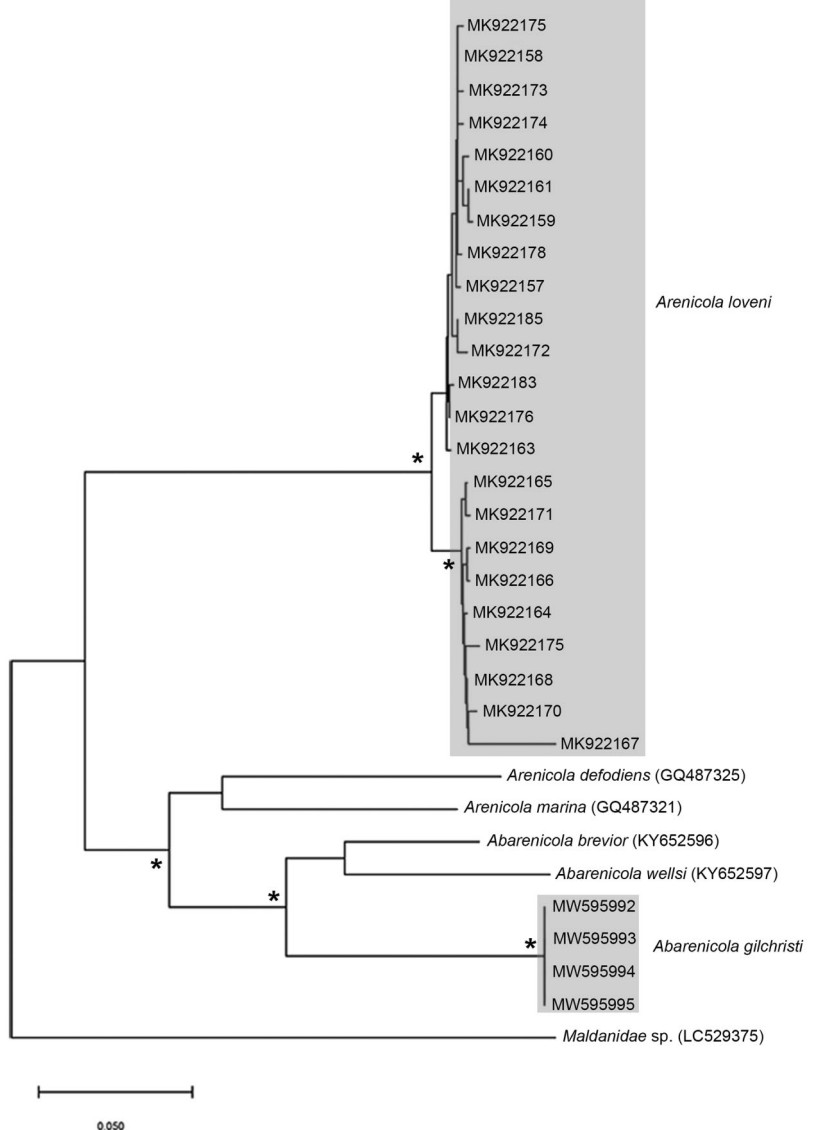

**Figure 7 Neighbour Joining tree using mitochondrial sequences belonging to various *Arenicola* Lamarck, 1801 and *Abarenicola* Wells, 1959 species, including *A. loveni* Kinberg, 1866 and *A. gilchristi* Wells, 1963 from South Africa.** *Indicates bootstrap support greater than or equal to 80%. Areas highlighted in grey represent sequences generated in this study. *Maldanidae* sp. was used as outgroup. Scale bar represents number of substitutions.

*Description*: Up to 89 mm long (excluding achaetous tail), 11 mm wide at chaetiger 1. In life, body orange-pink (Fig. 8A), light to dark pink when fixed (Fig. 8B). Epidermis tessellated to chaetiger 4 or middle of chaetiger 5, papillated thereafter. Chaetigerous annuli of first three chaetigers prominent, number of annuli between first four chaetigers 2-2 (3 in one specimen)-4, thereafter 4 (Fig. 8B). Anterior region consists of trilobed, non-retractable prostomium and one achaetous segment (Figs. 8B, 8D). Nuchal groove on each side (Fig. 8D). Proboscis eversible; covered in papillae, no pigment (Fig. 8C). Papillae on proximal section sparsely distributed, prominent, irregular in size, rounded (Fig. 8H). Papillae in

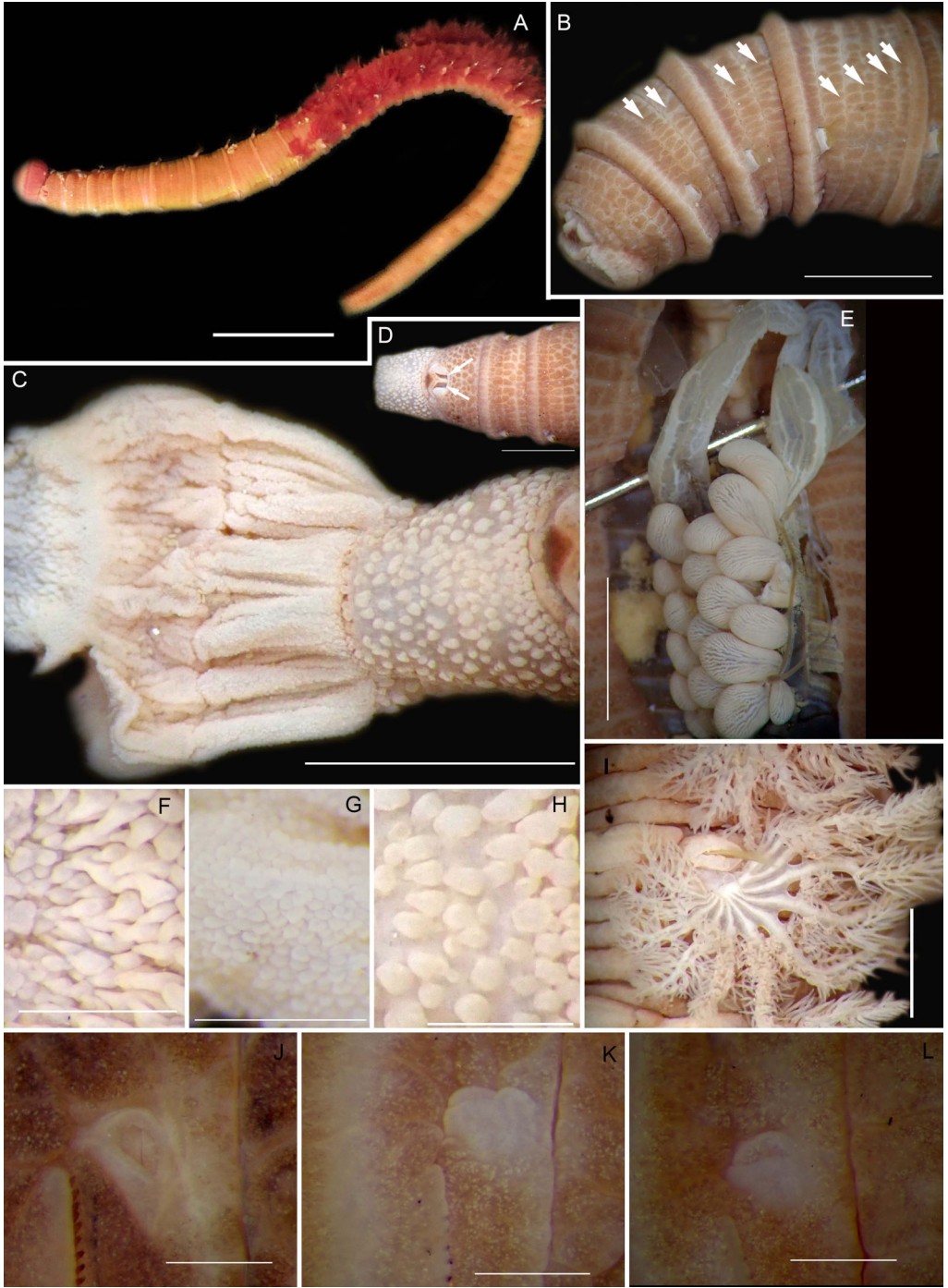

**Figure 8 Morphology of *Abarenicola gilchristi Wells, 1963*.** *(*A) Live specimen, (B) Dorso-lateral view of head, showing annulations on chaetigers 1 to 2 (arrowheads), (C) Proboscis showing papillations in different regions, (D) Dorsal view of head showing prostomium and partially everted proboscis, arrows show nuchal grooves, (E) Digestive caecae; one large pair and multiple smaller pairs, (F) Papillae of distal part of proboscis, (G) Papillae of median part of proboscis, (H) Papillae of proximal part of proboscis, (I) Branchia on chaetiger 9, (J) Unhooded nephridiopore, (K) Hooded nephridiopore, (L) Partially hooded nephridiopore. Scale bars: (A) = 2 mm, (B), (C), (D), (E) = 5 mm; (I) = 2 mm, (F)–(H) = 2 mm, (J)–(L) = 0.5 mm; (A), (J): MB-A090223, (B), (I), (K), (L) = MB-A090224; (C)–(H) = MB-A090226.

median section densely packed, small, rounded, skin folded (Fig. 8G). Papillae of distal section densely packed, conical (Fig. 8F). Oesophageal caecae with one elongate and 11 to 20 smaller caecae on either side of mid-line (Fig. 8E), elongate double to more than triple length of short caecae.

Thorax with 19 chaetigers. Notopodia rounded triangles, retractable lobes in oval torus. Notochaetae spinulose capillaries (Fig. 6C) in single row. Neuropodia oval bearing single row of unidentate, finely serrated, hooks (Fig. 6D). Neuropodia short, do not approach midline of venter. Branchiae on chaetigers 8–19 (12 pairs) (Fig. 8A). Branchiae highly vascularised, large, up to 19 main gill stems; highly branched, arborescent (tree-shaped), with lateral branches and gill filaments off each stem (Fig. 8I). Palmar membrane fuse lower third to half of gill stems (Fig. 8I). Five pairs of nephridia on chaetigers 5–9; nephridiopores unhooded, hooded, and partially hooded (Figs. 8J–8L), posterior to dorsal end of neuropodium. Tail achaetous, papillate, anus terminal.

*Remarks*: Specimens examined here conform to description by *Wells (1963)* and *Day (1967)*, but are smaller. *Abarenicola gilchristi* formed part of a distinct lineage in a well-supported clade (Fig. 7) also comprising *Abarenicola brevior* (*Wells, 1963*) and *A. wellsi Darbyshire, 2017*.

*Collection method*: By hand or digging with trowel.
*Type locality*: Buffelsbaai, Cape Peninsula, Western Cape Province, South Africa.
*Known distribution in South Africa*: Lambert's Bay to Walker Bay. Presence in Pearly Beach extends known distribution (*Day, 1967*) eastwards by only a few kilometres. Namibia: Luderitz. Report in Tamil Nadu, India (*Thilagavathi et al., 2013*) must be treated with caution.
*Ecology*: In sand in mid to low intertidal on sheltered shores.

Order: Sabellida *Levinsen, 1883*
Family: Sabellariidae *Johnston, 1865*
Genus: *Gunnarea Johansson, 1927*
Species: *Gunnarea gaimardi* (*de Quatrefages, 1848*)
Figure 9

*?Pallasia gaimardi de Quatrefages, 1848*: 24, 1866: 322, Pl. 13. Figs 17 & 18
*?Hermella capensis Schmarda, 1861*: 23, Pl. 23. Fig. 171. *?Sabellaria capensis McIntosh, 1885*: 418, Pl. 25A Figs 24 & 25, Pl. 26A Figs 11 712
*Gunnarea capensis Day, 1967*: Fig. 33.2.d-i (NOT *Schmarda, 1861*), *in partum*
*Gunnarea gaimardi Kirtley, 1994*: Fig. 3.1.2.a–e, *in partum*
*Gunnarea gaimardi Branch et al., 2016*: 73, Fig. 28.3
Common name: Coralworm, Cape reef worm, polwurm.

*Material examined:* Velddrif: 32°46′08.8″S 18°08′44.2″E, 10 specimens (incomplete), MB-A090356–MB-A090358, MB-A090360, MB-A090364, MB-A090367–MB-A090371, 26 May 2017, sand reefs in the mid-intertidal rock pools, coll. A. du Toit. Bettys Bay:

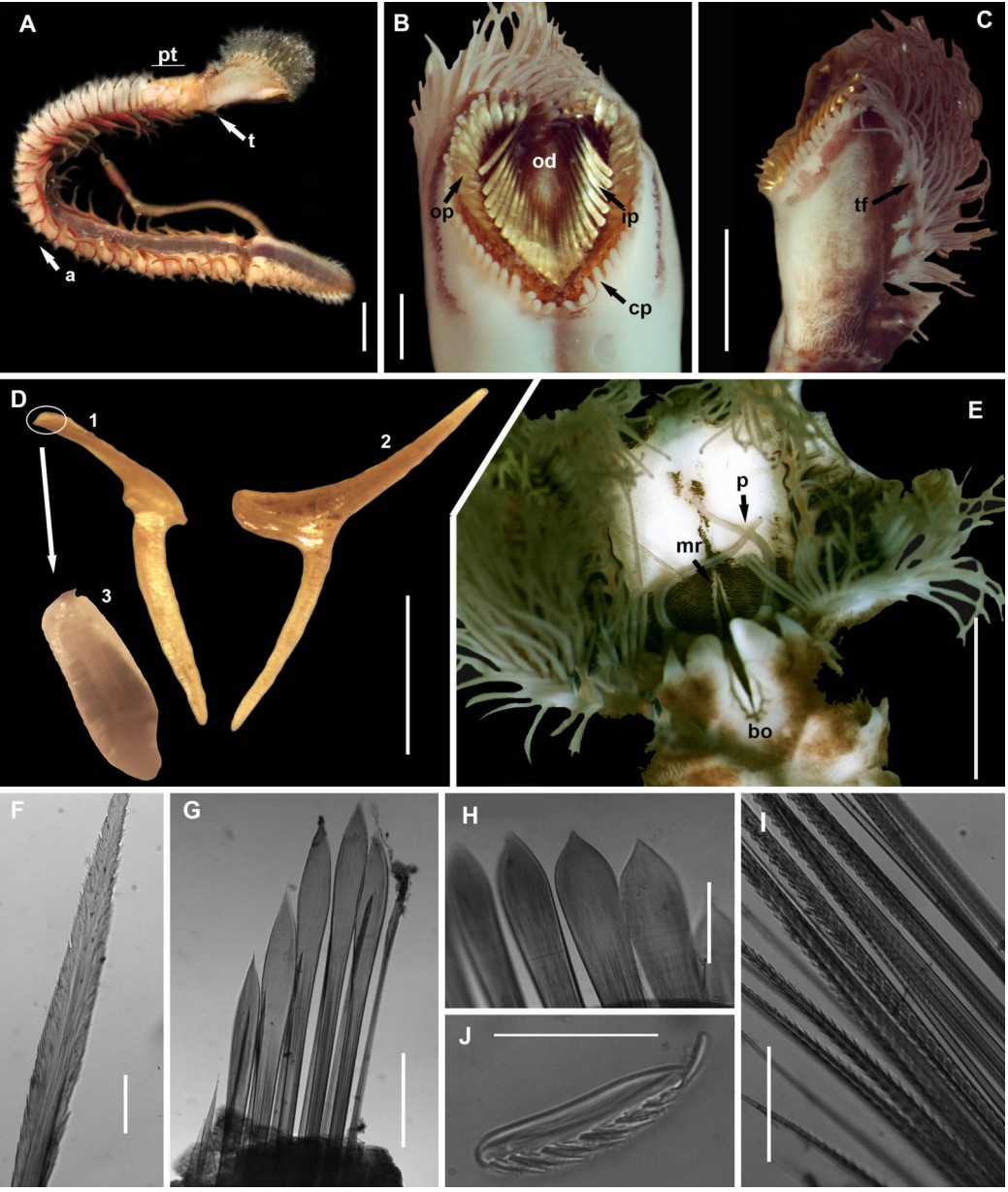

**Figure 9** **Morphology of *Gunnarea gaimardi* (*De Quatrefages, 1848*).** (A) Live specimen from Betty's Bay, (B) Dorsal view of crown showing the inner paleae (ip), outer paleae (op), opercular disk (od), papillae (cp), (C) Right side view of crown showing the tentacular filaments (tf), (D) Paleae, 1 & 3: outer geniculate paleae with tooth, 2: inner geniculate paleae, (E) Ventral view of anterior region showing palps (p), median ridge (mr), and U-shaped building organ (bo), (F) Bipinnate capillaries, neurochaetae, (G) Lanceolate chaetae of two lengths, neurochaetae, (H) Lanceolate and capillaries, notochaetae, (I) Verticillate chaetae, neurochaetae, (J) Uncini. Scale bars: (A) & (C) = 5 mm, (B) & (E) = 2 mm, (D) = 0.5 mm, (F), (I), (J) = 50 µm, (G) & (H) = 0.2 mm. (B), (F)–(J) = MB-A090337, (C) = MB-A090343, (D) = MB=A090371.

34°22′39.6″S 18°51′21.6″E, 5 specimens (incomplete), MB-A090336, MB-A090337, MB-A090339–MB-A090441), 3 June 2016, reefs in the lower intertidal zone, coll. E. Newman. Hermanus: 34°24′41.1″S 19°16′44.8″E, 8 specimens (incomplete),

MB-A090341–MB-A090348, 11 February 2017, low to mid intertidal, coll. A. du Toit and H. van Rensburg.

*Description:* Body a maximum of 110 mm in length; body colour opaque white and cream with irregular dark brown spots when fixed (Figs. 9A–9C). Opercular crown and opercular stalk completely fused (Fig. 9B). Two rows of golden outer and inner paleae, arranged in two concentric rows (Fig. 9B). Approximately 44–48 outer paleae and 35–46 inner paleae. Outer paleae geniculate, obtuse in shape with a single weak tooth on the antero-lateral margin (Figs. 9D1, 9D3). Inner paleae geniculate with elongate, wedge-shaped peaks with sharp tips (Fig. 9D2), arranged toward the midline of the crown with no overlap in paleae (Figs. 9B–9C). Anterior margin of crown with 49–73 conical papillae (Fig. 9B). Pair of ciliated palps in front of the mouth (Fig. 9E). Buccal lips present, with upper, lower and lateral lips (Fig. 9E). Tentacular filaments compound and branched (Fig. 9C). U-shaped building organ on the thorax (Fig. 9E); neurochaetae consists of capillaries with bipinnate blade margins (Fig. 9F); Parathorax consist of three chaetigers; notochaetae lanceolate interspersed with capillaries (Fig. 9H); neurochaetae alternating lanceolate chaetae of two lengths (Fig. 9G); neurochaetae thinner than notochaetae. Abdomen with pairs of branchiae on each segment; neuropodial lobes reduced on abdominal chaetigers, surrounded by tori; uncini with five teeth (Fig. 9J); neurochaetae verticillate chaetae (Fig. 9I); ventral cirri conical with tapering ends, becoming digitiform with rounded ends, spanning the neuropodial lobe.

*Remarks:* Specimens collected from all western sites (Veldrif, Betty's Bay and Hermanus) conformed to the general descriptions according to *Day (1967)* and *Kirtley (1994)*. Nonetheless, differences in the morphology of the outer paleae were observed. *Day (1967)* described two incurving teeth present on the outer paleae, but this differs from what was observed in specimens collected in the present study: one tooth on the antero-lateral margin of the outer paleae. Additionally, *Day (1967)* described the inner paleae as completely concealing the "fleshy disk" or opercular disk, however, this was not observed for our specimens, instead the opercular disk was visible in the mid-section where paleae did not overlap, which was similar to *Kirtley (1994)*. Nonetheless, all other characters observed for our specimens were similar to specimens as described by *Kirtley (1994)* and *Day (1967)*, suggesting that they most likely represent *Gunnarea gaimardi sensu stricto*. Molecular analyses (Fig. 10) will be discussed under *Gunnarea* sp. 1.

*Collection method:* Breaking off pieces of reef by hand or narrow blade to remove worms from tubes.
*Type locality*: Cape of Good Hope, Western Cape Province, South Africa.
*Known distribution in South Africa*: the nominal species has been reported from KwaZulu-Natal on the east coast to the west coast of the Western Cape Province; Namibia: Walvis Bay to Luderitz (*Day, 1967*), but this needs to be revised.
*Ecology:* Species forms extensive reefs by building sandy tubes on rocks in the low to mid intertidal of exposed shores.

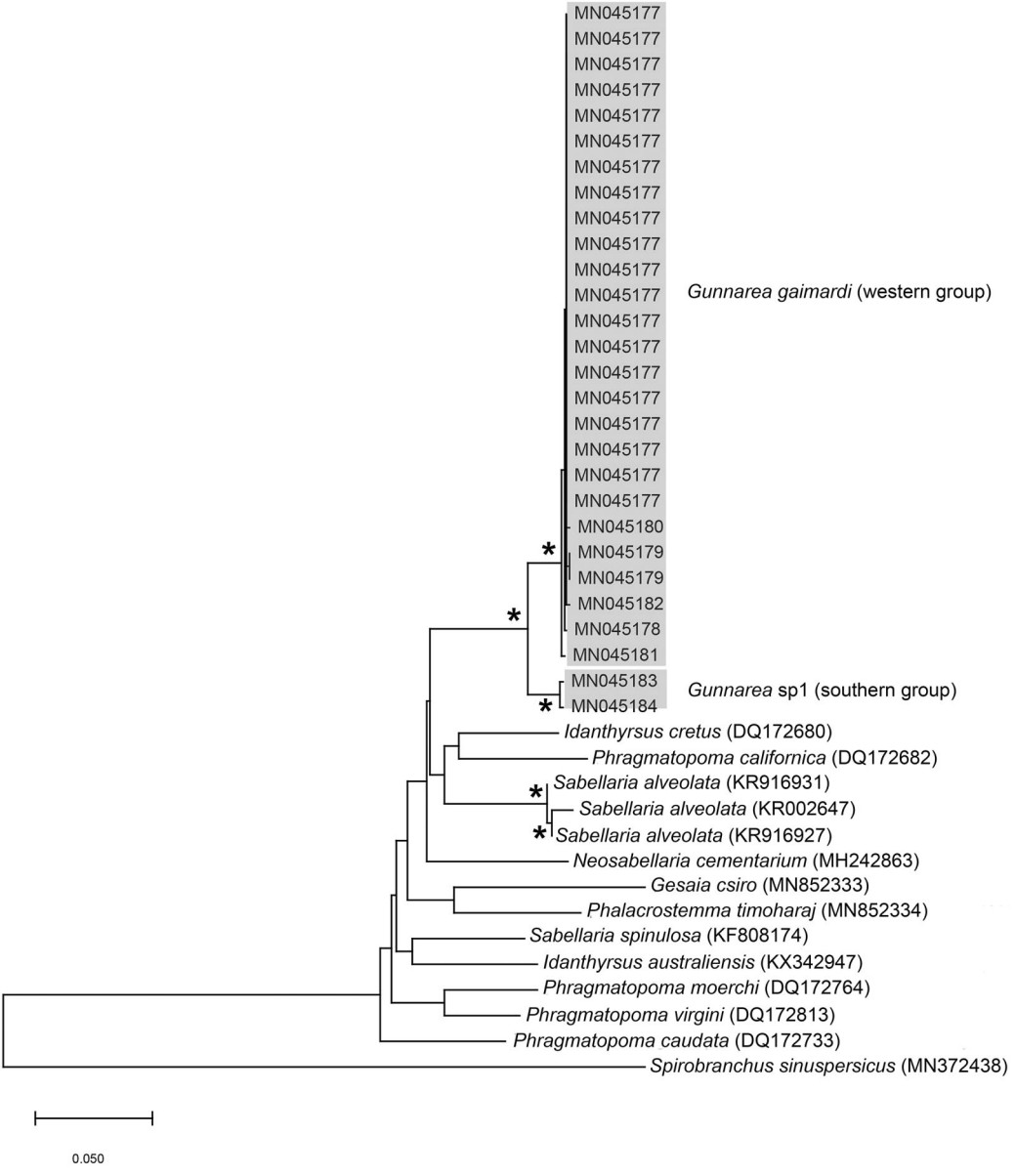

MN045177
MN045177
MN045177
MN045177
MN045177
MN045177
MN045177
MN045177
MN045177
MN045177
MN045177
MN045177
MN045177
MN045177
MN045177
MN045177
MN045177
MN045177
MN045177
MN045180
MN045179
MN045179
MN045182
MN045178
MN045181
MN045183
MN045184

*Gunnarea gaimardi* (western group)

*Gunnarea* sp1 (southern group)

*Idanthyrsus cretus* (DQ172680)
*Phragmatopoma californica* (DQ172682)
*Sabellaria alveolata* (KR916931)
*Sabellaria alveolata* (KR002647)
*Sabellaria alveolata* (KR916927)
*Neosabellaria cementarium* (MH242863)
*Gesaia csiro* (MN852333)
*Phalacrostemma timoharaj* (MN852334)
*Sabellaria spinulosa* (KF808174)
*Idanthyrsus australiensis* (KX342947)
*Phragmatopoma moerchi* (DQ172764)
*Phragmatopoma virgini* (DQ172813)
*Phragmatopoma caudata* (DQ172733)
*Spirobranchus sinuspersicus* (MN372438)

0.050

**Figure 10 Neighbour Joining tree of mitochondrial sequences of various species from family Sabellariidae** *Johnston, 1865* **including** *Gunnarea capensis* **(***Schmarda, 1861***).** *Indicates bootstrap support greater than 80%. Areas in grey represent sequences generated in this study. *Spirobranchus sinuspersicus Pazoki et al., 2020* was used to root the tree. Scale bar represents number of substitutions.

Species: *Gunnarea* sp. 1
Figure 11

Material examined: Witsand: 34°23′31.9″S 20°51′50.1″E, 2 specimens (incomplete), MB-A090293, MB-A090294, 30 April 2017, low to mid intertidal, coll. A. du Toit.

*Description*: Body maximum of 43 mm (MB-A090293) in length (MB-A090294 = 34 mm), when fixed body colour opaque white with black pigmentation throughout (Figs. 11A,

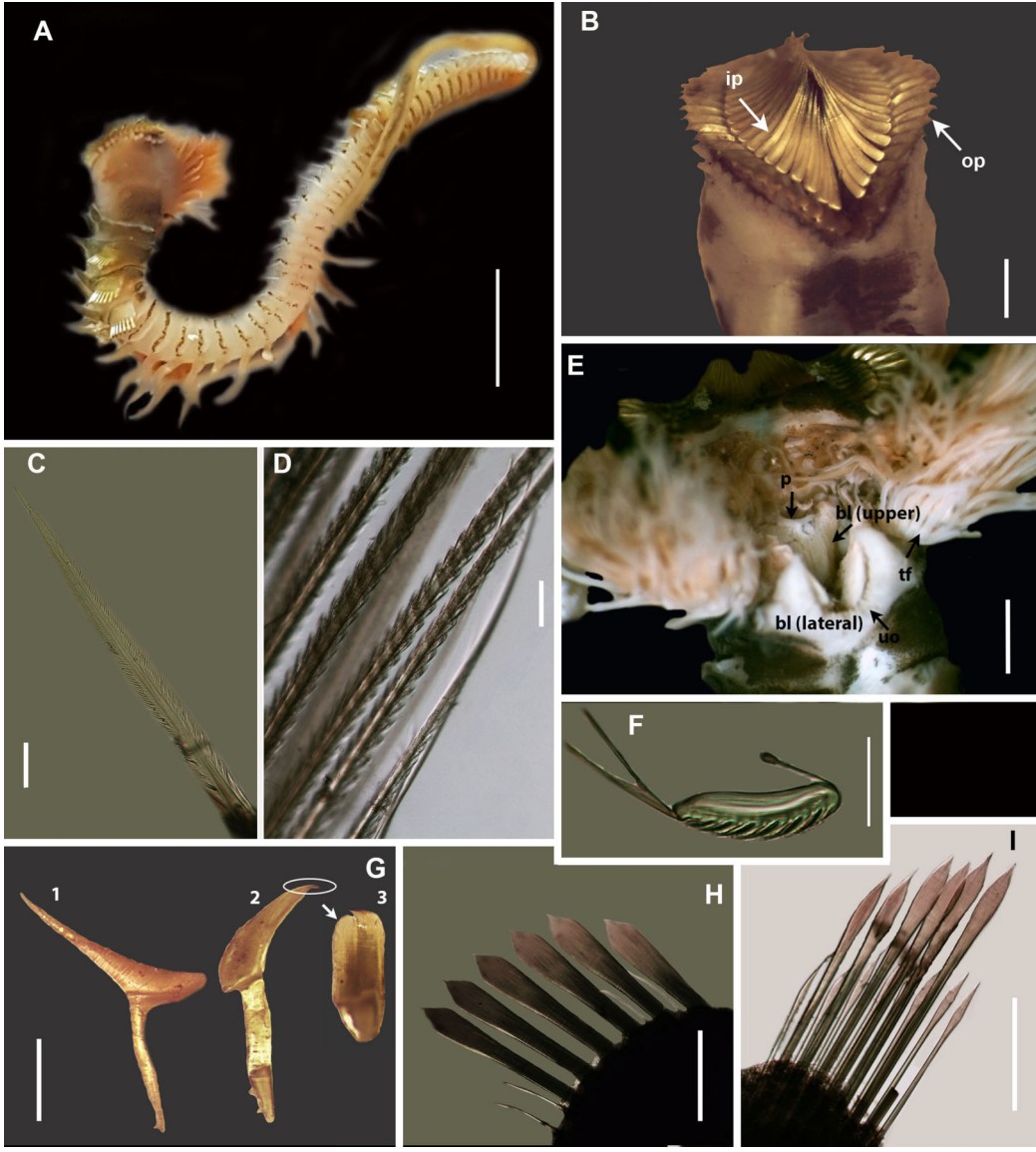

**Figure 11 Morphology of *Gunnarea Johansson, 1927* species collected from Witsand.** (A) Live specimen, (B) Crown showing the inner paleae (ip) and outer paleae (op), (C) Neurochaetae of first thoracic chaetiger, (D) Abdominal neurochaetae, (E) Anterior region showing palps (p), buccal lip (bl, upper and lateral sides) and tentacular filaments (tf), (F) Posterior uncinus, (G) Palaea 1- inner geniculate paleae, 2 & 3 - outer geniculate paleae with tooth, (H) Lanceolate notochaetae, (I) Neurochaetae. Scale bars: (A) = 5 mm, (B) & (E) = 1 mm, (C), (D), (F), (G) = 0.5 mm, (H)–(I) = 0.2 mm. (A), (C), (F)–(I) = MB-A090293, (B), (E) = MB-A090294.

11B). Opercular crown and opercular stalk completely fused (Fig. 11B). Two rows of golden inner and outer paleae (Fig. 11B). Approximately 34–38 inner paleae and 42–43 outer paleae. Outer paleae geniculate with a single tooth on the antero-dorsal margin (Figs. 11G2, 11G3), inner paleae geniculate with elongate, wedge-shaped peaks with a sharp tip (Fig. 11G1), and orientated toward the midline, with both rows overlapping and concealing the opercular disk (Fig. 11B). Anterior margin of opercular crown with 50 conical papillae (Fig. 11B). Pair of ciliated palps in front of mouth (Fig. 11E). Buccal lips

present with upper, lower and lateral lips (Fig. 11E). Tentacular filaments compound and branched (Fig. 11E). U-shaped building organ as part of thorax (Fig. 11E); neurochaetae capillaries with bipinnate blade margins (Fig. 11C). Parathorax of three chaetigers; notochaetae alternating lanceolate and capillary chaetae (Fig. 11H), neurochaetae lanceolate chaetae of two lengths (Fig. 11I), neurochaetae thinner than notochaetae. Abdomen with a pair of branchiae on each segment; reduced neuropodial lobes surrounded by tori, uncini with seven teeth (Fig. 11F); neurochaetae verticillate (Fig. 11D). Ventral cirri conical with tapering ends, becoming digitiform with rounded ends spanning the neuriopodial lobes.

*Remarks:* Specimens collected from Witsand (southern site) conformed to the general description by *Day (1967)*, including having paleae that completely conceal the "fleshy disk" or opercular disk. Specimens from the southern site generally resemble *Gunnarea gaimardi* (from western sites) in having a single tooth on the antero-lateral margin of the outer paleae. Nonetheless several differences were observed. Firstly, western site specimens were longer (max. of 110 mm), whereas southern specimens were a maximum of 43 mm. The most distinct feature between these two morpho-groups was the shape, orientation and arrangement of paleae on the opercular crown. The peaks of the outer and inner paleae are longer in specimens from the southern site compared to that observed in specimens from the western sites; the angle of inclination between the handle and peaks of the inner paleae is larger in western specimens than southern specimens; the outer paleae blades are wider and shorter in specimens from southern sites compared to the longer, thinner blades observed in western specimens. The inner paleae in western specimens do not overlap at the midpoint of the opercular disk, thereby exposing the disk, whereas in southern specimens the paleae overlap, completely concealing the disk. Additionally, the abdominal uncini of western specimens have five teeth, which is two less than that observed for southern specimens. Lastly, western specimens have more opercular papillae than southern specimens when comparing similar sized animals; 73, length 45 mm and 50, length 43 mm, respectively. These differences noted between specimens collected from the southern site and *G. gaimardi* from western sites indicate that they are indeed separate species and that specimens from the southern site (Witsand) most likely represents a new undescribed species of the genus. These morphological differences are supported by the molecular analysis which recovered two well supported clades (Fig. 10) and a genetic distance of 6% (±0.02), thus confirming their separation as independent species. The first clade, designated *G. gaimardi*, included specimens from Velddrif, Betty's Bay and Hermanus (western group) and the second, designated *Gunnarea* sp. 1, included only the specimens from Witsand (Fig. 10). Morphological differences together with the genetic separation of the clades indicate the presence of two species in what has, till now, been considered a monospecific genus (*Capa, Hutchings & Peart, 2012*). Preliminary observations of *Gunnarea* sp. from Port Shepstone in KwaZulu-Natal suggest that they conform to the description of *Gunnarea* sp. 1 and studies are underway to confirm this.

*Collection method:* Breaking off pieces of reef by hand or narrow blade to remove worms from tubes.

*Known distribution in South Africa*: Witsand, Western Cape Province, South Africa.

*Ecology:* Species forms extensive reefs by building sandy tubes on rocks in the low to mid intertidal of exposed shores.

Order: Eunicida *Dales, 1962*
Family: Lumbrineridae *Schmarda, 1861*
Genus: *Scoletoma Blainville, 1828*
Species: *Scoletoma* sp. 1
Figure 12

?*Lumbrinereis tetraurus Day, 1953*: 435
?*Lumbrineris tetraura Day, 1967*: 437, 439, Fig. 17.16 U–W, *Branch et al., 2016*: 70, Fig. 26.10
Common name: Puddingworm.

*Material examined:* Betty's Bay: 34°22′S 18°51′E, 1 specimen (incomplete), MB-A090332, 3 June 2016, sandy sediment, coll. E. Newman.

*Description*: more than 145 mm; L10 = 8.1 mm, W10 = 3.4 mm (Fig. 12A). Prostomium conical, peristomium with two rings, second slightly shorter than first (Fig. 12A). No eyes. Prechaetal lobes truncate throughout, postchaetal lobe longer and bluntly triangular, becoming longer towards posterior end (Figs. 12C–12E). Winged capillary chaetae from chaetiger 1 to approximately chaetiger 57. Long-headed simple multidentate hooded hooks (about 0.2 mm long) from approximately chaetiger 4, shortening posteriorly (Figs. 12F, 12G), after about chaetiger 35, head becomes even shorter with flared hood (Fig. 12H), appearing white. Aciculae yellow. Dental formula: MI = 1 + 1, MII = 5 + 5, MIII = 2 + 2, MIV = 1 + 1 (Fig. 12B), MV free, lateral to MIV and MIII.

*Remarks:* The specimen conforms to the general description *of S. tetraura* according to *Schmarda (1861)* and *Day (1967)*, and no characters could be identified to distinguish the specimen collected here from the description of specimens from Chile. However, the specimen differs morphologically and genetically from others collected in this study that also conform to the description of *S. tetraura* by *Day (1967)*, and genetically from specimens identified as *S. tetraura* in China (Fig. 13; discussed under *Scoletoma* sp. 2 below). The species is therefore identified here as *Scoletoma* sp. 1.

*Collection method*: Collected with a small trowel from sediment.

*Apparent distribution in South Africa*: Known only from a single site. It closely resembles *Scoletoma tetraura* which has been reported from Namibia to KwaZulu-Natal.

*Ecology:* Burrows into sand in rock pools.

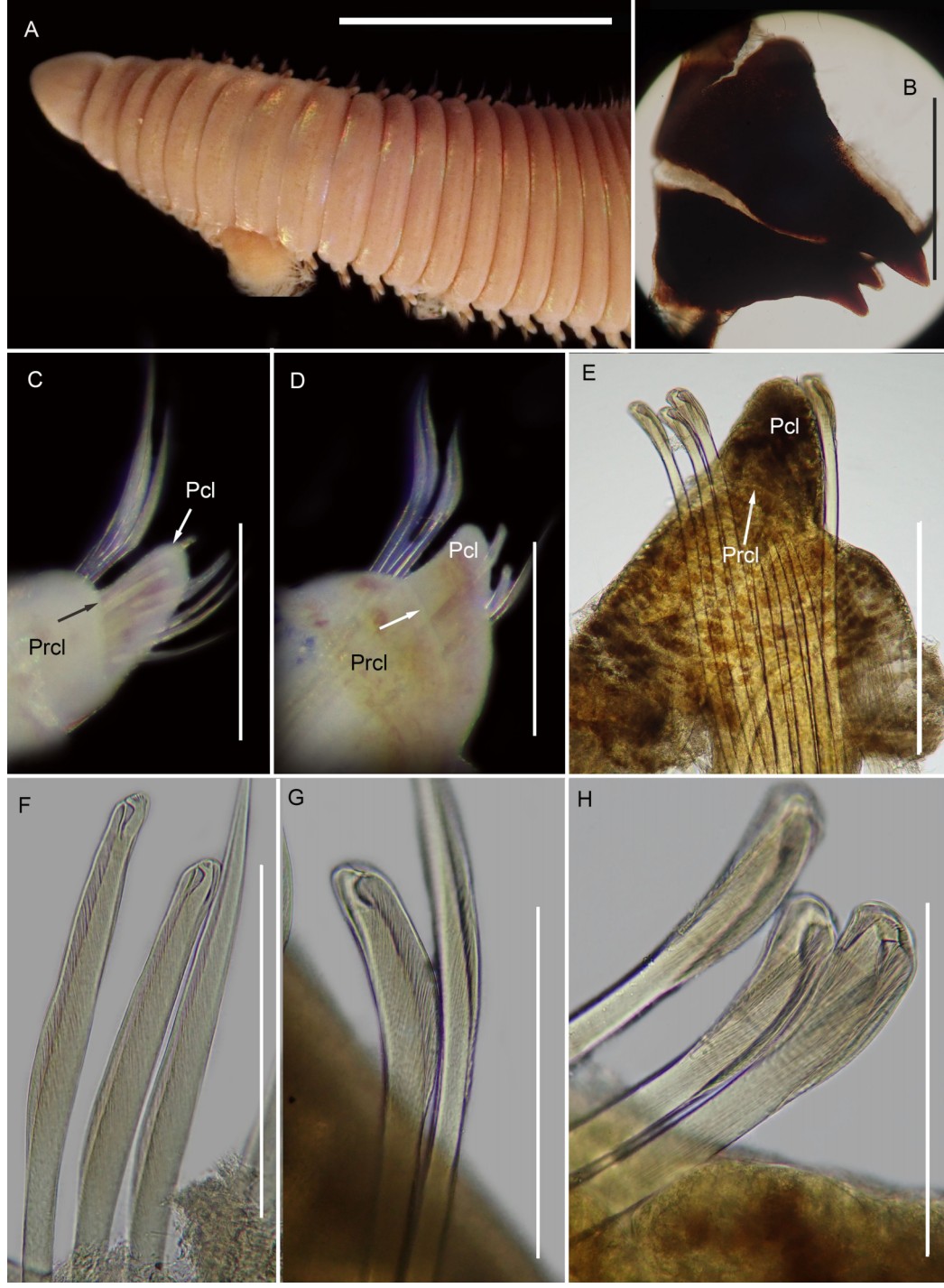

**Figure 12 Morphology of *Scoletoma* species 1 from Betty's Bay.** (A) Dorsal anterior, (B) MIII and MIV of jaws, ventral view, (C) Chaetiger 5 showing pre-chaetal (Prcl) and post chaetal (Pcl) lobes, anterior view, (D) Chaetiger 31 showing prechaetal and post chaetal lobes, anterior view, (E) Posterior chaetiger showing prechaetal and post chaetal lobes, anterior view, (F) Long-headed multidentate hooded hooks on chaetiger 5, (G) Long-headed multidentate hooded hook on chaetiger 31, (H) Short-headed multidentate hook with flared hood from posterior chaetiger. Scale bars: (A) = 5 mm, (B)–(E) = 0.5 mm, (F)–(H) = 0.2 mm. (A)–(H) = MB-A090332.

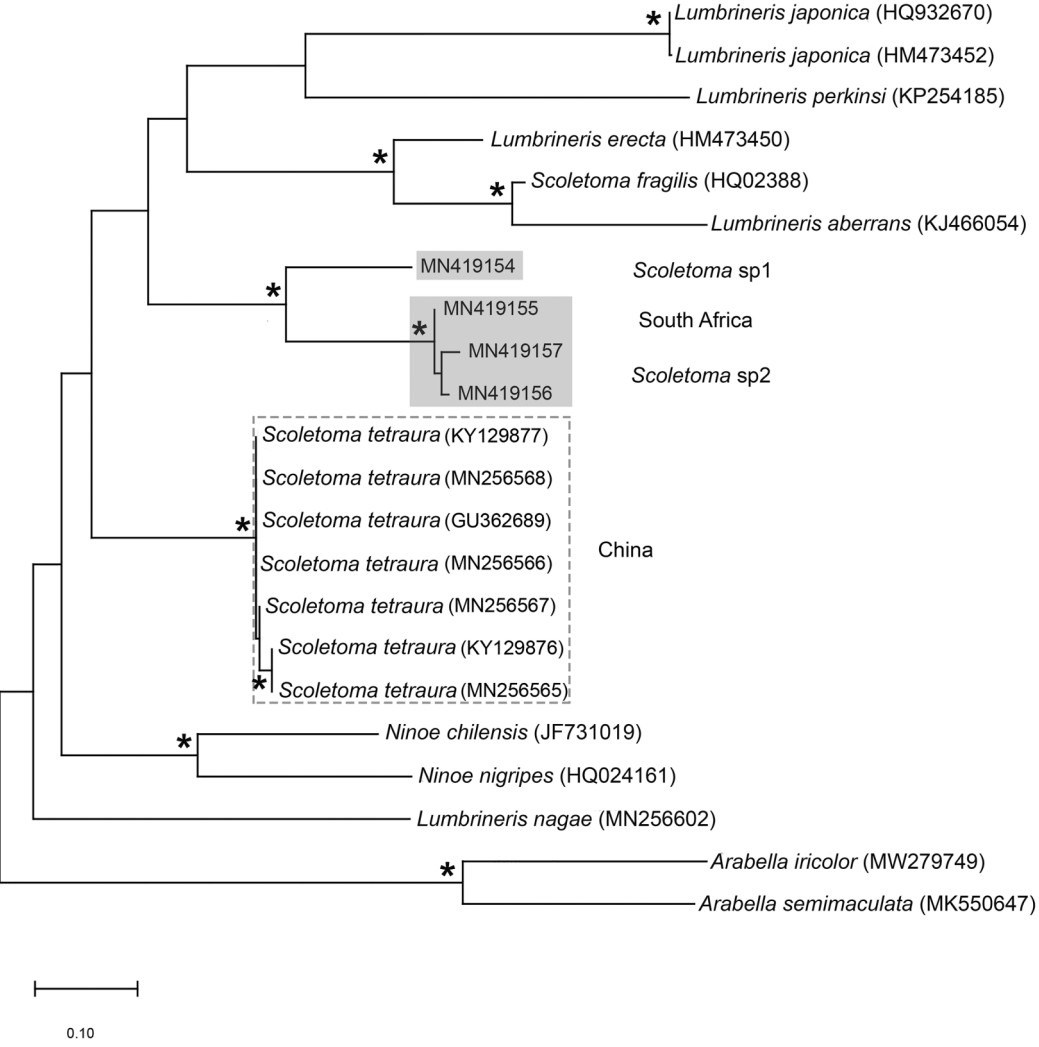

**Figure 13 Neighbour joining tree of mitochondrial sequences of various species in Lumbrineridae, including *Scoletoma tetraura* (*Schmarda, 1861*) from China.** * Indicates bootstrap support greater than 80%. Areas highlighted in grey represent sequences generated in this study; *Scoletoma* species sp. 1 and sp. 2. Area outlined with grey dashed line represents *S. tetraura* from China. *Arabella iricolor* (*Montagu, 1804*) and *A. semimaculata* (*Moore, 1911*) were used as outgroups. Scale bar indicates number of substitutions.

Species: *Scoletoma* sp. 2
Figure 14

?*Lumbrinereis tetraurus* *Day, 1953*: 435
?*Lumbrineris tetraura* *Day, 1967*: 437, 439, Fig. 17.16 u–w, *Branch et al., 2016*: 70, Fig. 26.10
Common name: Puddingworm.

*Material examined:* Hermanus, Kammabaai: 34°24′41.1″S 19°16′44.8″E, 6 specimens (incomplete), MB-A090349–MB-A090354, 11 February 2017, from rock pools in low to mid intertidal, coll. A. du Toit and H. van Rensburg.

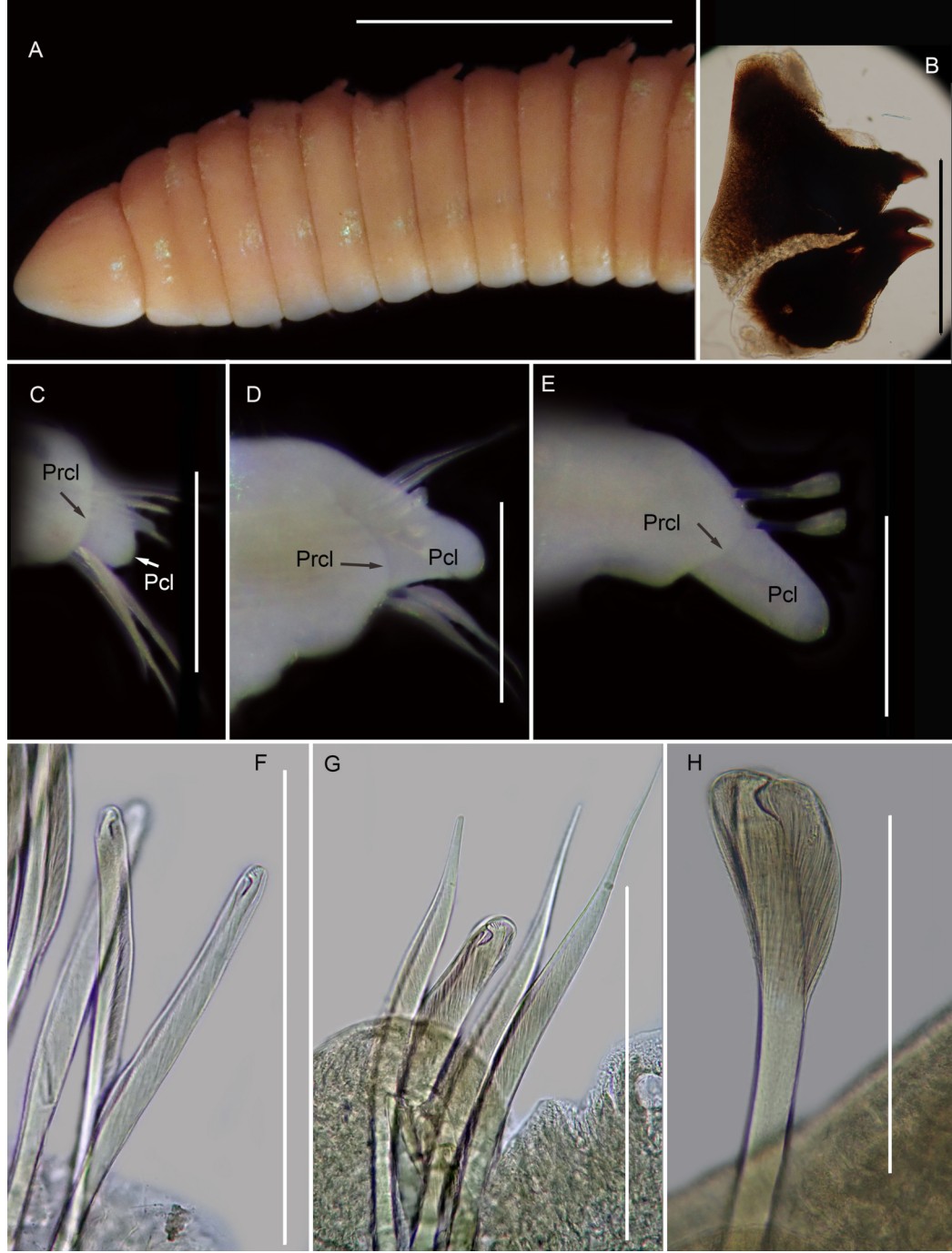

**Figure 14 Morphology of *Scoletoma* species 2 from Hermanus.** (A) Dorsal anterior, (B) MIII and MIV of jaws, ventral view, (C) Chaetiger 3 showing pre-chaetal (Prcl) and post chaetal (Pcl) lobes anterior view, (D) Chaetiger 30 showing pre- and post chaetal lobes, anterior view, (E) Posterior chaetiger showing pre- and post chaetal lobes, anterior view, (F) Long-headed multidentate hooded hooks on chaetiger 5, (G) Long-headed multidentate hooded hook on chaetiger 31, (H) Short-headed multidentate hook with flared hood from posterior chaetiger. Scale bars: (A) = 5 mm, (B)–(E) = 0.5 mm, (F)–(H) = 0.2 mm. (A)–(H) = MB-A090353.

*Description*: up to more than 300 mm; L10 = 6.8 to 9.8 mm, W10 = 1.9 to 3.4 mm. Prostomium conical, peristomium with two rings, second slightly shorter than first (Fig. 14A). No eyes. Prechaetal lobes truncate, short and rounded throughout, postchaetal lobe longer and bluntly triangular in anterior chaetigers, becoming digitiform and longer towards posterior end (Figs. 14C–14E). Winged capillary chaetae from chaetiger 1 to approximately chaetiger 56 to 70. Long-headed simple multidentate hooded hooks (about 0.15mm long) from approximately chaetiger 4, shortening posteriorly (Figs. 14F, 14G), after about chaetiger 25, but usually after about chaetiger 30 to 35, head becomes even shorter with flared hood (Fig. 14H), appearing white (Fig. 14E). Aciculae yellow. Dental formula (variation): MI = 1 +1, MII = 5 (6) + 5, MIII = 2 (1) + 1 (2), MIV = 1 + 1 (Fig. 14B), MV free, lateral to MIV and MIII.

*Remarks*: All six specimens conform to the general description of *S. tetraura* according to Schmarda (1861) and Day (1967), and no characters could be identified to distinguish the specimens collected here from the description of specimens from Chile. However, this species differs from the specimen from Betty's Bay. In *Scoletoma* sp. 2 from Hermanus, the long-headed simple hooded hooks are about 25% shorter than those of *Scoletoma* sp. 1 from Betty's Bay, and post-chaetal lobes are about 30% longer in the posterior. Furthermore, the segments of *Scoletoma* sp. 2 appear to be longer than those of *Scoletoma* sp. 1; in specimens that are similarly wide, specimens of the former are 1.5 to 1.8 mm longer for the first 10 chaetigers than in the latter. Finally, specimens of the two species were collected from different habitats. Further research is needed to determine which, if any, refers to the species recorded previously by Day (1967) as *S. tetraura*.

The morphological separation is supported by molecular analyses (Fig. 13) that retrieved two well-supported operational taxonomic units, *Scoletoma* sp. 1 (from Betty's Bay) and *Scoletoma* sp. 2 (from Hermanus). The two *Scoletoma* species from South Africa form part of a weakly supported clade together with *Scoletoma fragilis* (O.F. Müller, 1776), *Lumbrineris aberrans* Day, 1963, *Lumbrineris erecta* Moore, 1904, *Lumbrineris japonica* Marenzeller, 1879, and *Lumbrineris perkinsi* Carrera-Parra, 2001 which is separate from *S. tetraura* from China. The separation of *Scoletoma* spp 1 and 2 from South Africa and *S. tetraura* from China in two different clades with high support suggests that they are independent species. However, without sequences from the species' type locality in Chile, it is impossible to determine whether the specimens found in China and South Africa all represent new species or whether one of them is an alien. Specimens from the extended global distribution *of S. tetraura* need to be examined, as there are likely more species within this complex. Additionally, *S. tetraura* and *S. fragilis* were previously considered members of *Lumbrineris*, so the other *Lumbrineris* species in the clade should be revised to determine whether they are also in the genus *Scoletoma*, or whether this genus is paraphyletic.

*Collection method*: Samples from Hermanus collected among broken pieces of *Gunnarea* tubes.

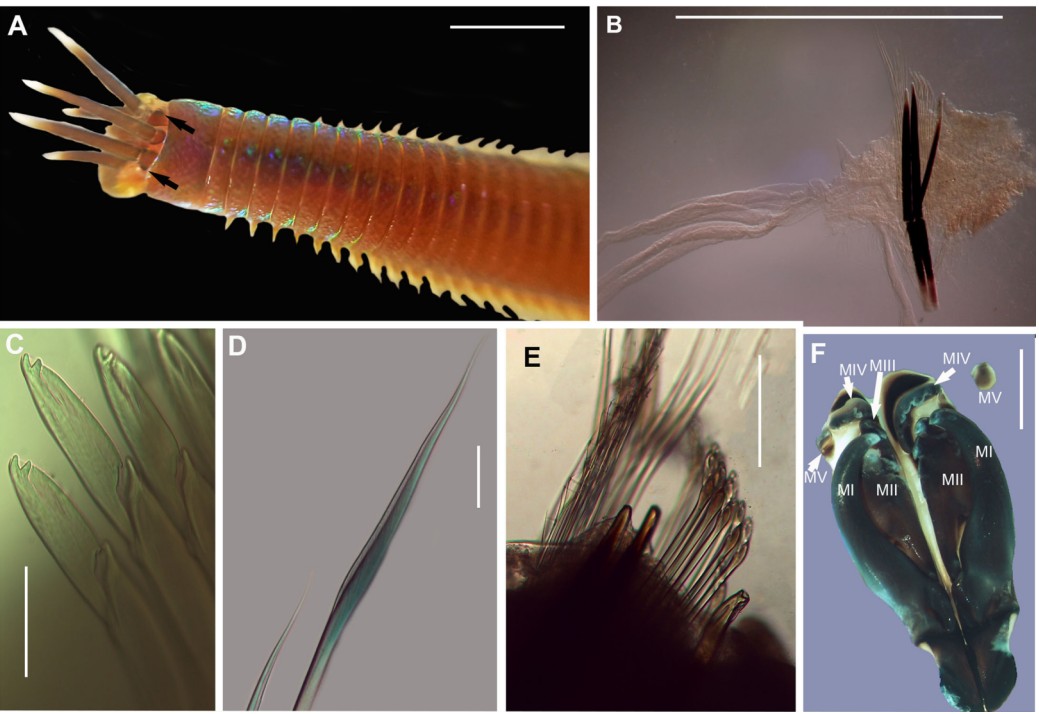

**Figure 15 Morphology of *Marphysa* cf. *corallina*.** (A) Anterior region (dorsal), live specimen, (B) Posterior chaetiger with branchia, (C) Compound bidentate falcigers with guards, (D) Limbate capillaries, (E) Chaetae, acicula and subacicular hooks, (F) Dorsal view of Maxillary apparatus Scale bars: (A) = 5 mm, (B) = 0.5 mm, (C) & (D) = 50 μm, (E) = 0.2 mm, (F) = 2 mm. (A), (C) & (D) = MB-A090276, (E) & (F) = MB-A090280.     

*Apparent distribution in South Africa*: Known only from a single site. It closely resembles *Scoletoma tetraura*, which has been reported from Namibia to KwaZulu-Natal.
*Ecology*: Burrows into sand among *Gunnarea* tubes.

Family: Eunicidae *Berthold, 1827*
Genus: *Marphysa de Quatrefages, 1866*
Species: *Marphysa* cf. *corallina*
Figure 15

*Marphysa corallina Day, 1967*: 400, Fig. 17.7 F–J; *Branch et al., 2016*: 70, Fig 26.7
Common name: Wonderworm.

*Material examined*: Witsand: 34°23′31.9″S 20°51′50.1″E, 5 specimens, (incomplete) MB-A090276–MB-A090280, 30 April 2017, under rocks in rock pools in mid-intertidal, coll. A. du Toit.

*Description*: Body length more than 120 mm; L10 = 8–11 mm, W10 = 0.4–0.5 mm. In live specimens, body colour medium to dark brown in anterior becoming light brown in posterior; iridescent throughout (Fig. 15A). Prostomium bilobed, lobes frontally rounded; sulcus deep. Prostomial appendages semi-circular with white tapering tips (Fig. 15A);

pair of palps extend to second peristomial ring; pair of lateral antennae reaching second segment and one median antenna extending to third segment (Fig. 15A). Black reniform eye spots below pair of lateral antennae (Fig. 15A, black arrows). Four pairs of maxillary plates and one maxilla; MI = 1 + 1, MII = 3 + 3, MIII = 5 + 0, MIV = 4 + 6, MV = 1 + 1 (Fig. 15F). Branchiae pectinate, from chaetiger 35–47 onwards present as a single filament, reaching up to five to seven filaments in middle chaetigers (Fig. 15B). Dorsal cirri digitform in anterior, middle and posterior chaetigers. Ventral cirri conical in anterior chaetigers and reduces to an oval swelling with a rounded tip in posterior chaetigers. Aciculae blunt with dark brown tips and black shafts (Figs. 15B, 15E); 3 per fascicle in anterior segments, reducing to 2 and then 1 in middle segments; subacicular hooks, light brown tips with black shafts, present from 40th chaetiger with bidentate tips and guards (Figs. 15B, 15E). Limbate capillaries present in supracicular fascicle throughout (Figs. 15D, 15E). Pectinate chaetae present in supracicular fascicle; isodont broad blades and fine teeth (Fig. 15E). Compound falcigers, bidentate tips, short blades with guards, present in subacicular fascicle (Fig. 15C).

*Remarks*: Specimens collected in this study conform to the general description according to *Day (1967)*. Unfortunately, the original description of *M. corallina* (*Kinberg, 1865a*) was poor, with no illustrations against which to compare the specimens collected in this study. However, since the type locality of *M. corallina* is in Hawaii and the species has a global disjunct distribution, it is probable that the specimens collected here are really an incorrectly identified indigenous species. We therefore take the more conservative route and refer to the species collected in South Africa as *M.* cf. *corallina*. All specimens collected during this study were incomplete, missing their posterior ends, so characters such as anodont chaetae, the number of branchial filaments and the number of aciculae in the posterior regions were not documented and thus could not be commented on.

All sequences generated clustered with *M. corallina* from KwaZulu-Natal (KT823410) (*Kara, 2015*), with high bootstrap support, indicating that it is a single species (Fig. 16). Further investigation is underway to confirm the taxonomic status of *M. corallina* in South Africa.

*Collection method*: By hand from sediment under rocks.
*Known distribution in South Africa*: Mabibi in northern KwaZulu-Natal to Mgazana in the Eastern Cape Province, Witsand in Western Cape Province (*Day, 1967*; current study).
*Apparent distribution globally*: Mozambique, New Zealand, Red Sea, Australia, Marshall Islands, Lakshadweep Island and Juluit Atoll (*Day, 1967*; *Read & Fauchald, 2021*).
*Ecology*: Occupies burrows in sediment under rocks in the mid-intertidal zone.

Species: *Marphysa haemasoma de Quatrefages, 1866*
Figure 17

*Marphysa haemasoma de Quatrefages, 1866*: 334–334, Figs. 4B, 6 & 7; Grube 1870: 299
*Marphysa sanguinea Day, 1967*: 396, fig. 17.5 U–Y (NOT Montagu, 1815)

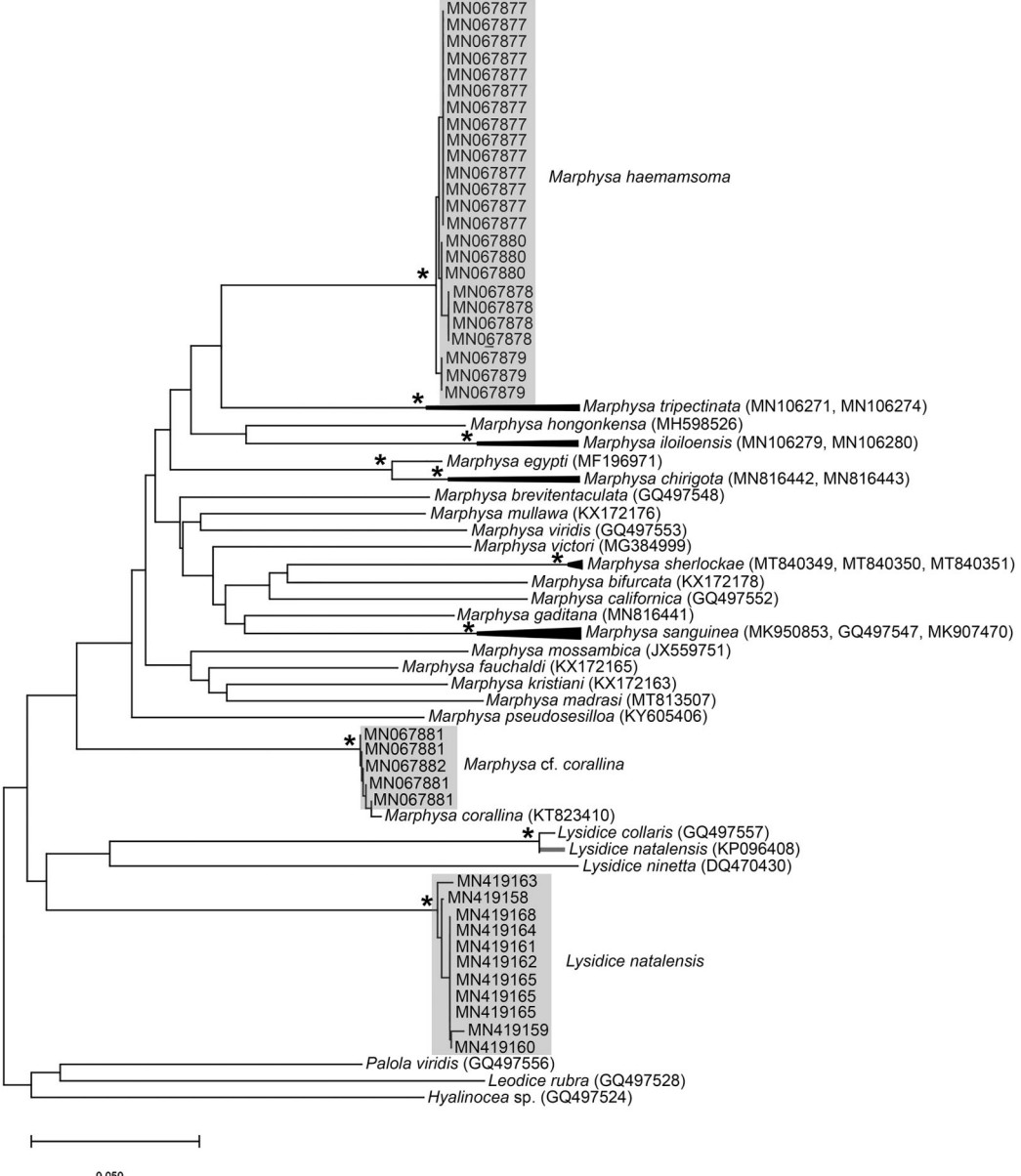

**Figure 16 Neighbour Joining tree of various species belonging to family Eunicidae *Berthold, 1827*, including *Marphysa De Quatrefages, 1866* and *Lysidice Lamarck, 1818* from South Africa.** * Indicates bootstrap support greater than or equal to 80%. Grey highlighted areas indicate sequences generated in this study. Red branch represents a questionable sequence labelled as *Lysidice natalensis* Kinberg, 1865 from India. *Palola viridis* Gray in Stail, 1847, *Leodice rubra Grube, 1856* and *Hyalinocea* sp. were used as outgroups. Scale bar indicates number of substitutions.

*Marphysa elityeni Lewis & Karageorgopoulos, 2008*: 280–281, Figs. 1 &2; *Branch et al., 2016*: 69, Fig. 2.5

*Marphysa haemasoma Kara et al., 2020*: 16–21, Figs 4B, 6 & 7

Common name: Wonderworm, bloukoppies. Listed as estuarine wonderworm in *Branch et al. (2016)*.

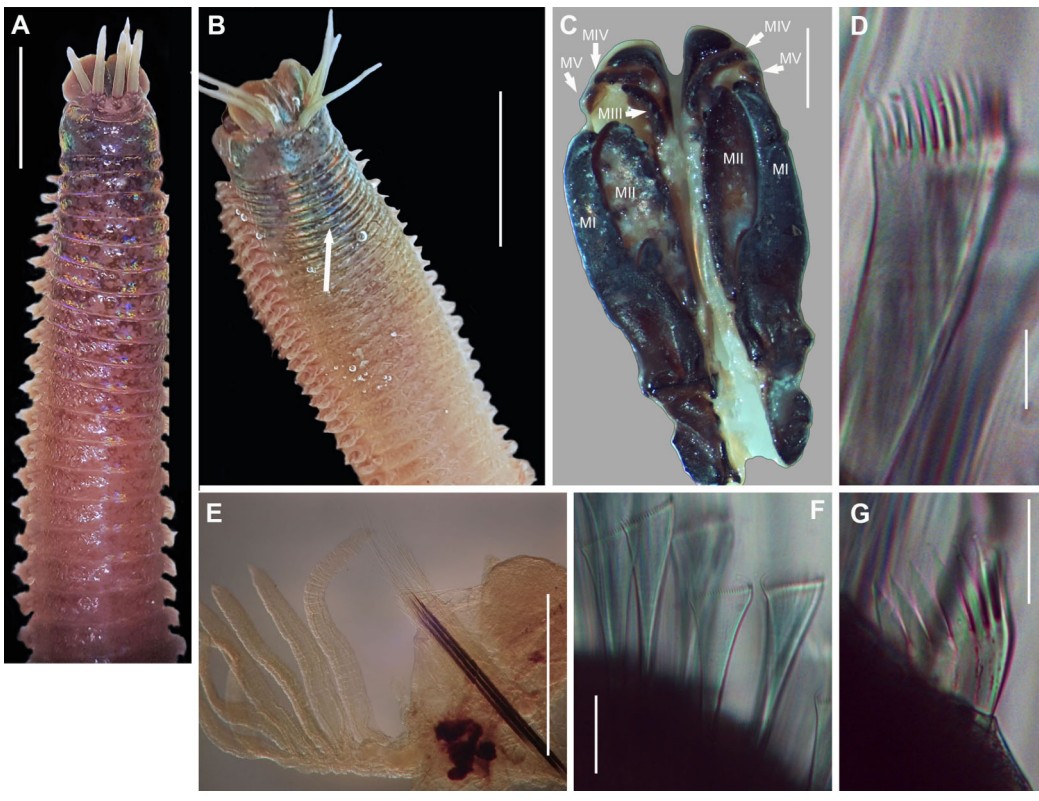

**Figure 17 Morphology of *Marphysa haemasoma* *De Quatrefages, 1866*.** (A) Dorsal anterior, live specimen, (B) Dorsal anterior of live specimen, arrow showing blue colouration, (C) Dorsal view of maxillary apparatus, (D) Pectinate anodont chaetae, (E) Middle chaetiger with branchia, (F) Pectinate isodont chaetae, (G) Pectinate anodont chaetae. Scale bars: (A) & (B) = 5 mm, (C) = 2 mm, (D), (F) & (G) = 50 μm, (E) = 0.25 mm. (A) = MB-A090326, (B) = MB-A090328, (C) = MB-A090274, (D), (F)–(G) = MB-A090273.

*Material examined:* Knysna: 34°02′17.5″S 23°02′23.4″E, 2 specimens (incomplete), MB-A090326, MB-A090328, 29 January 2017, coll. A. du Toit. Betty's Bay: 34°22′S 18°51′E, 5 specimens (incomplete), MB-A090331, MB-A090333–MB-A090335, MB-A090338, 3 June 2016, digging with a trowel in mid-intertidal rock pools, coll. E. Newman. Strand: 34°07′03.2″S 18°49′29.4″E, 2 specimens, MB-A090271, MB-A090315, 13 January 2017, digging with trowel in gravel under rocks in the mid-intertidal, coll. A. du Toit. Soetwater: 34°09′33.0″S 18°19′40.7″E, 5 specimens (incomplete specimens), MB-A090272–MB-A090275, MB-A090317, 10 March 2017, under rocks in mid-intertidal rock pools, coll. A. du Toit. Melkbosstrand: 33°43′40.3″S 18°26′17.6″E, 4 specimens (incomplete), MB-A090267–MB-A090270, 26 February 2017, under rocks in mid-intertidal rocky reef, coll. A. du Toit and C. Naidoo.

*Description*: Body length more than 470mm. In life body colour variable: dark brown/red anterior with white iridescent spots for about 7 chaetigers (Fig. 17A), becoming medium brown in middle and darker towards the posterior. Specimens from Knysna and Betty's Bay with blue colouration in anterior for about 6 chaetigers (Fig. 17B, white arrow), becoming light brown in middle to posterior. Body iridescent in all specimens.

Prostomium bilobed, lobes frontally rounded, sulcus deep (Figs. 17A, 17B). Prostomial appendages in semi-circle with a brown band just before the tapering ends in live specimens; pair of palps reaching first peristomial ring, pair of lateral antennae extending to second segment and one median antenna reaching first chaetiger (Figs. 17A, 17B). Pair of eyes under the lateral antennae. Four pairs of maxillary plates and a maxilla (variation); MI = 1 + 1, MII = 3 (4) + 4, MIII = 5 + 0, MIV = 3 + 5, MV = 1 + 1 (Fig. 17C). Branchiae pectinate, present from chaetiger 26 onwards as two filaments, reaching a maximum of 8 filaments in middle, reducing to a single filament in middle to posterior, absent in posterior end near pygidium. Acicula black (Fig. 17E) throughout, five per fascicle in anterior chaetigers, reducing to three in posterior; subacicular hooks not observed. Simple capillaries and pectinate chaetae present in supracicular fascicle. Four types of pectinate chaetae; isodonts with fine teeth and symmetrical blades (Fig. 17F) in anterior segments and anodonts with medium and coarse teeth (Figs. 17D, 17G) in middle to posterior chaetigers. Compound spinigers with short and long blades present in subacicular fascicle throughout.

*Remarks*: Specimens collected here conformed to the description by *Kara et al. (2020)*, except for those collected from Knysna and Betty's Bay which have a blue anterior (approximately first six chaetigers), becoming light brown in the middle to posterior end. Phylogenetic analysis recovered a single well-supported clade that comprised all specimens from Knysna, Betty's Bay, Strand, Kommetjie (Soetwater) and Melkbosstrand, indicating that the colour morphs are a single species (Fig. 16). *Lewis & Karageorgopoulos (2008)* observed colour variation in specimens which included iridescent blues and greens for the reproductive segments along the length of the body, from chaetigers 70–80. This does not conform to the colour morphs found in the specimens in the present study in which the colour was observed in the anterior regions. Nonetheless, the colour on the remaining parts of the body, "medium brown in the middle and darker toward the posterior", conform to that reported by *Lewis & Karageorgopoulos (2008)*.

The use of two species of *Marphysa* in the Western Cape Province supports recent research showing that globally, multiple species of this genus, especially members of the *M. sanguinea* complex, are used as bait, even within regions (see review by *Hutchings & Lavesque, 2021*). Although the current study showed that different colour morphs represent a single species, further research is needed to determine whether individuals occupying different habitats, as described by *Day (1967)* and *Lewis & Karageorgopoulos (2008)*, are also a single species.

*Collection method*: By hand from sediment under boulders in boulder fields.
*Type locality*: Cape of Good Hope, Western Cape, South Africa.
*Known distribution in South Africa*: Langebaan Lagoon on the west coast to Port Elizabeth on the south coast (*Day, 1967*; *Kara et al., 2020*).
*Ecology*: Occupies burrows in sediment typically grey/black medium to coarse grains and rich in sulphur. In Knysna, specimens were found in sandier sediments.

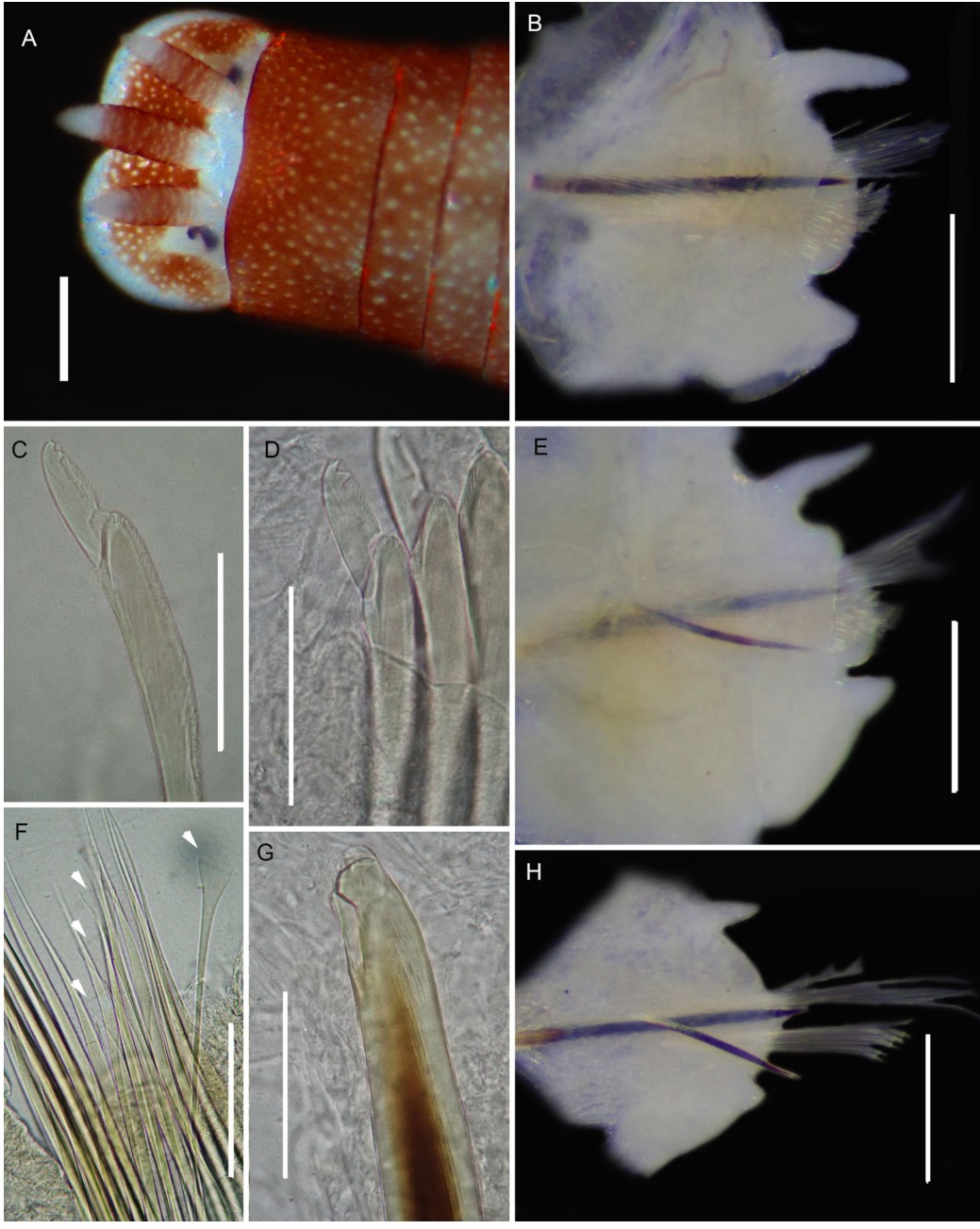

**Figure 18 Morphology of *Lysidice natalensis Kinberg, 1865a*.** (A) Dorsal of head, live specimen, (B) Chaetiger 5, anterior view, (C) Compound falciger of chaetiger 5, (D) Compound falciger of chaetiger 28, (E) Chaetiger 28, anterior view (F) Limbate and comb (white arrowheads) chaetae of chaetiger 28, (G) Acicula hook of chaetiger 28, (H) Posterior chaetiger. Scale bars: (A) = 1 mm, (B), (E), (H) = 0.5 mm, (C), (D), (F), (G) = 0.05 mm. (A)–(H) = MB-A090291.

Genus: *Lysidice Lamarck, 1818*
Species: *Lysidice natalensis Kinberg, 1865a*
Figure 18

*Lysidice natalensis Kinberg, 1865a*: 566; *Hartman, 1948*: 84, 85, Pl. XI Figs. 1–2; *Day, 1951*: 40; *Day, 1953*: 435; *Day, 1960*: p 336; *Day, 1967*: 401, Fig. 17.7 K–R; *Branch et al., 2016*: 70, Fig. 26.9
*Lysidice atra Schmarda, 1861*
*Lysidice capensis* Grube, 1868: 12, Fig. 4; *Day, 1934*: 53
Common name: Musselworm. Listed as three-antennaed worm in *Branch et al. (2016)*.

*Material examined*: Witsand: 34°23′31.9″S 20°51′50.1″E, 11 specimens (2 complete), MB-A090281–MB-A090289, MB-A090291, MB-A090292, 30 April 2017, from under rocks, in rock pools in mid-intertidal zone, coll. A. du Toit.

*Description*: Complete specimens 62 and 63 mm long for 126 and 156 chaetigers. L10 5.28–9.8 mm, W10 1.84–4.5 mm. Colour reddish-brown with white spots, both extending into middle of prostomium and antennae, margin of prostomium and tips and base of antennae white (Fig. 18A). Prostomium bilobed, antennae tapered, lateral antennae shorter than prostomium, median antenna slightly longer, proximal part brown, tips white (Fig. 18A). Mandibles thick; MI 1 + 1; MII 3 + 3; MIII 2-3 + 0; MIV 2-3 +4-7; MV 1 + 1. Parapodia with slender dorsal cirri (Fig. 18B), becoming shorter and thinner from chaetiger 22 to 38 onwards (Figs. 18E, 18H). Ventral cirrus bluntly triangular (Fig. 18B), getting shorter posteriorly (Fig. 18E), nipple-shaped in posteriormost chaetigers (Fig. 18H). Post-chaetal lobe truncate (Fig. 18B), getting shorter posteriorly (Fig. 18E), inconspicuous in posteriormost chaetigers (Fig. 18H). Superior chaetae limbate capillaries and comb chaetae of two sizes (Fig. 18F). Inferior compound chaetae with short blades, bidentate, teeth usually of similar sizes (Figs. 18C, 18D), but proximal tooth may be thicker and or longer. Acicula black with blunt tips, one in anterior chaetigers, two in middle and posterior (Figs. 18B, 18E, 18H); bidentate acicula hook with small hood from chaetiger 25–28 onwards (Figs. 18E, 18H), teeth may be worn, giving unidentate appearance (Fig. 18G).

*Remarks*: Original description by *Kinberg (1865a)* is poor, but this material is later described by *Hartman (1948)*. Specimens collected here generally match this latter description, and those by *Day (1951, 1953, 1967)*, although the posterior ventral cirrus is more prominent than described by *Day (1967)*. The wide distribution within South Africa is suggestive of multiple species and may be further reflected by the two species that *Day (1967)* synonymised with *L. natalensis* without explanation. It is therefore possible that *L. capensis* and *L. atra*, both originally described from the temperate Western Cape Province in Kalk Bay and the Cape of Good Hope, respectively, are not *L. natalensis* which was first described from Durban in the subtropical KwaZulu-Natal. Additionally, *Day (1967)* provides no explanation for why *L. atra*, which was described four years before *L. natalensis* and therefore claims priority, was synonymised with the latter. More specimens from throughout the distribution range and any available type material need to be examined to resolve the taxonomy of this species. The description of *L. natalensis* from Pakistan by *Mustaquim (2000)* is not very detailed, and the only differences from samples examined here are differently shaped post-chaetal lobes. All specimens from

Witsand form a well-supported clade that is not reciprocally monophyletic with *L. natalensis* from India (Fig. 16; *Sigamani et al., 2020*). Identity of the species in Pakistan is also doubtful.

*Collection method*: By hand.
*Type locality*: Durban, KwaZulu-Natal, South Africa.
*Known distribution in South Africa*: From Namibia to northern KwaZulu-Natal (*Day, 1967*).
*Ecology*: Habitat variable; in the current study specimens were collected from under rocks in rock pools, *Day (1934)* reported them from muddy sand.

Family: Onuphidae *Kinberg, 1865a*
Genus: *Heptaceras Ehlers, 1868*
Species: *Heptaceras quinquedens* (*Day, 1951*)
Figure 19

*Onuphis quinquedens Day, 1951*: 40–42, Fig. 6A–H; *Day, 1967*: 422, Fig. 17.13A–E; *Fauchald, 1982*: 100, Fig. 28B
*Heptaceras quinquedens Paxton, 1986*: 58–60, Fig. 36I, J
Common name: moonshineworm.

*Material examined*: Pearly Beach: 34°40′00.5″S 19°29′42.7″E, 5 specimens (incomplete), MB-A090432–MB-A090436, 23 January 2017, coll. H. van Rensburg & A. du Toit. Strand beach: 34°06′37.6″S 18°49′14.6″E, 1 specimen (incomplete), MB-A090442, 13 January 2017, coll. H. van Rensburg and A. du Toit. Struisbaai Main Beach: 34°47′32.3″S 20°02′54.8″E, 15 specimens (incomplete), MB-A090421–MB-A090431, MB-A090437–MB-A090440, 27 January 2017, coll. H. van Rensburg, A. du Toit and C. Naidoo.

*Description*: Large species reaching 350 mm in length and 6mm width at $10^{th}$ chaetiger. Anterior section rounded, becoming dorso-ventrally flattened and ventrally convex from chaetiger 3–6 onward (Fig. 19D). In life, prostomium and peristomium white (Fig. 19B), rest of body pale, white-brown ventrally and more reddish-brown dorsally (Fig. 19E), becoming paler towards median and posterior sections, dorsum covered with small red-brown spots, more prominent towards anterior (Fig. 19B). Irregularly spaced red-brown or black dots on ceratophoral rings with a single white patch within final elongated distal ring (Fig. 19B). All colouring disappears after preservation (Figs. 19A, 19C, 19D). Iridescent shine observed over entire body in live and preserved specimens (Figs. 19A–19D).

Prostomium with frontal extension forming palpohores for frontal palps (Fig. 19C). Lateral antennae reaching chaetiger 4–7 on posterior part of prostomium, shorter median antenna reaching chaetiger 2–4 placed anterior to lateral antennae. Proximal ceratophoral rings wide, covering most of prostomium (Fig. 19A). Ceratophores with 15–30 rings on median antennae and 20–48 rings on lateral antennae, each terminating in an elongated distal ring. Ceratophores at least as long as styles but up to twice the length of styles which taper distally (Figs. 19B, 19D). Peristomium as long as, or longer than, prostomium with

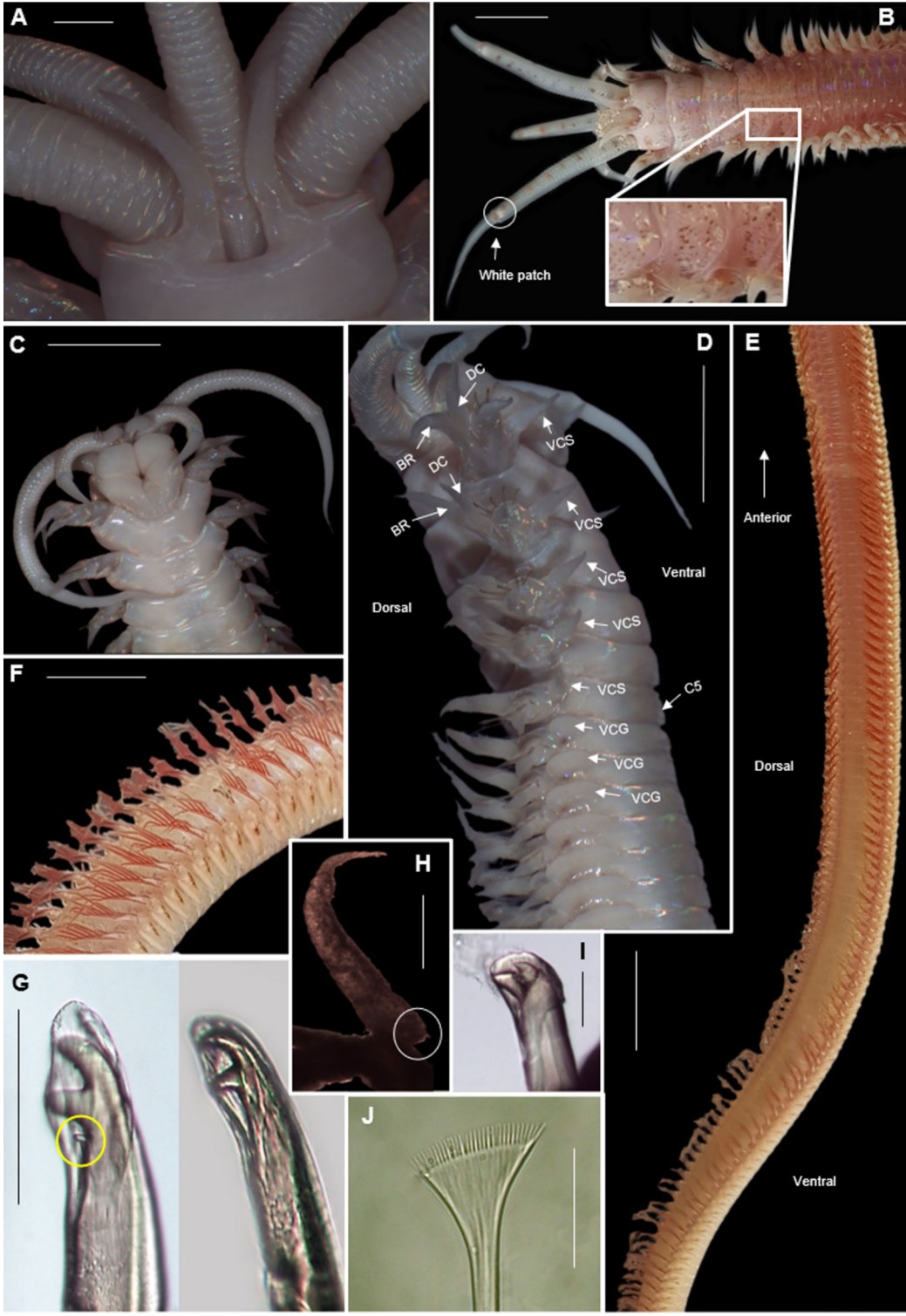

**Figure 19 Morphology of *Heptaceras quinquedens* (*Day, 1951*).** (A) Dorsal anterior of preserved specimen showing peristomial notch flanking prostomial ridge, laterally curving peristomial cirri and iridescent shine that remains after preservation, (B) Dorsal anterior of live specimen, insert shows freckled spots on anterior dorsum, (C) Ventral anterior of preserved specimen, (D) Lateral anterior view of preserved specimen showing cylindrical shape of modified parapodia and progression of

**Figure 19 (continued)**
ventral cirri from subulate to globular form, (E) Dorsal view of live specimen from chaetiger 11–92
showing fading of colouration from anterior to middle of body, (F) Lateral view of live mid-section, (G)
Bidentate and tridentate falcigers, with minor third tooth encircled, (H) Dorsal cirri from chaetiger 82
with small basal process encircled, (I) Bidentate acicular chaetae, (J) Pectinate chaetae. DC = Dorsal cirri,
BR = Branchiae, VCS = Ventral cirri subulate form, VCG = Ventral cirri globular form, C5 = Chaetiger
five. Scale bars: (A) = 1 mm; (B), (C), (D), (F) = 5 mm; (E) = 10 mm; (G), (I) = 0.1 mm; (J) = 5 μm. (A),
(D), (H) = MB-A090434; (B), (E), (F) = MB-A090442; (C), (G), (J) = MB-A090424.

deep mid-dorsal notch on the dorsal margin, flanking an elevated prostomial ridge
(Fig. 19A). Peristomial cirri as long as peristomium, slender and tapering, situated distally
on peristomium on either side of the mid-dorsal notch, curving laterally (Figs. 19A, 19B).

   Parapodia mounted marginally, anterior three pairs projecting anteriorly, slightly
elongated (Figs. 19B, 19D) and modified with four or five hooded bi- or tridentate
pseudo-compound falcigers (Fig. 19G), remaining parapodia directed dorsally. Dorsal cirri
simple tapering filament anteriorly with small basal process towards posterior end
(Fig. 19H), shorter than branchiae (Figs. 19D, 19F). Ventral cirri subulate on anterior five
chaetigers changing to pad-like globular form (Fig. 19D). Pectinate chaetae from chaetiger
6–8 with 22–28 teeth (Fig. 19J). Superior limbate chaetae from chaetiger 1. Branchiae
start as simple tapered filaments on chaetiger 1 (Fig. 19D), become pectinate on chaetiger
8–10 with maximum of 7–12 filaments per branchia (Fig. 19F), continuing throughout rest
of body (Fig. 19E). Hooded bidentate acicular chaetae appear from 10$^{th}$ chaetiger to the
end of the body (Fig. 19I).

*Remarks*: The specimens examined here match earlier descriptions (*Day, 1951*, *1967*;
*Fauchald, 1982*), but this is the first observation of tridentate falcigers in the modified
parapodia, although tridentate falcigers are known to occur within the genus (*Paxton,
1986*). The third tooth is small (Fig. 19G) and not always present so can easily be
overlooked. According to *Fauchald (1982)* the median antenna is longer than the posterior
lateral ones in the holotype (reaching chaetiger three vs. two) but in all of the material
examined here, the posterior lateral antennae were longer than the median antenna,
conforming to the description by *Paxton (1986)*. The iridescent shine seen on the body of
*H. quinquedens* is similar to that of *Diopatra aciculata* (*Van Rensburg, Matthee & Simon,
2020*) and may be why fishermen commonly refer to both species as moonshineworms.

*Collection method:* "prawn pumps" during low tide.
*Type locality:* Umpangazi, KwaZulu-Natal, South Africa.
*Known distribution in South Africa:* Western Cape Province to KwaZulu-Natal
(*Day, 1967*).
*Apparent distribution globally:* report in India (*Sigamani et al., 2020*) needs to be
confirmed.
*Ecology:* They build temporary tubes in the intertidal of sandy beaches, but do not build
conspicuous chimneys.

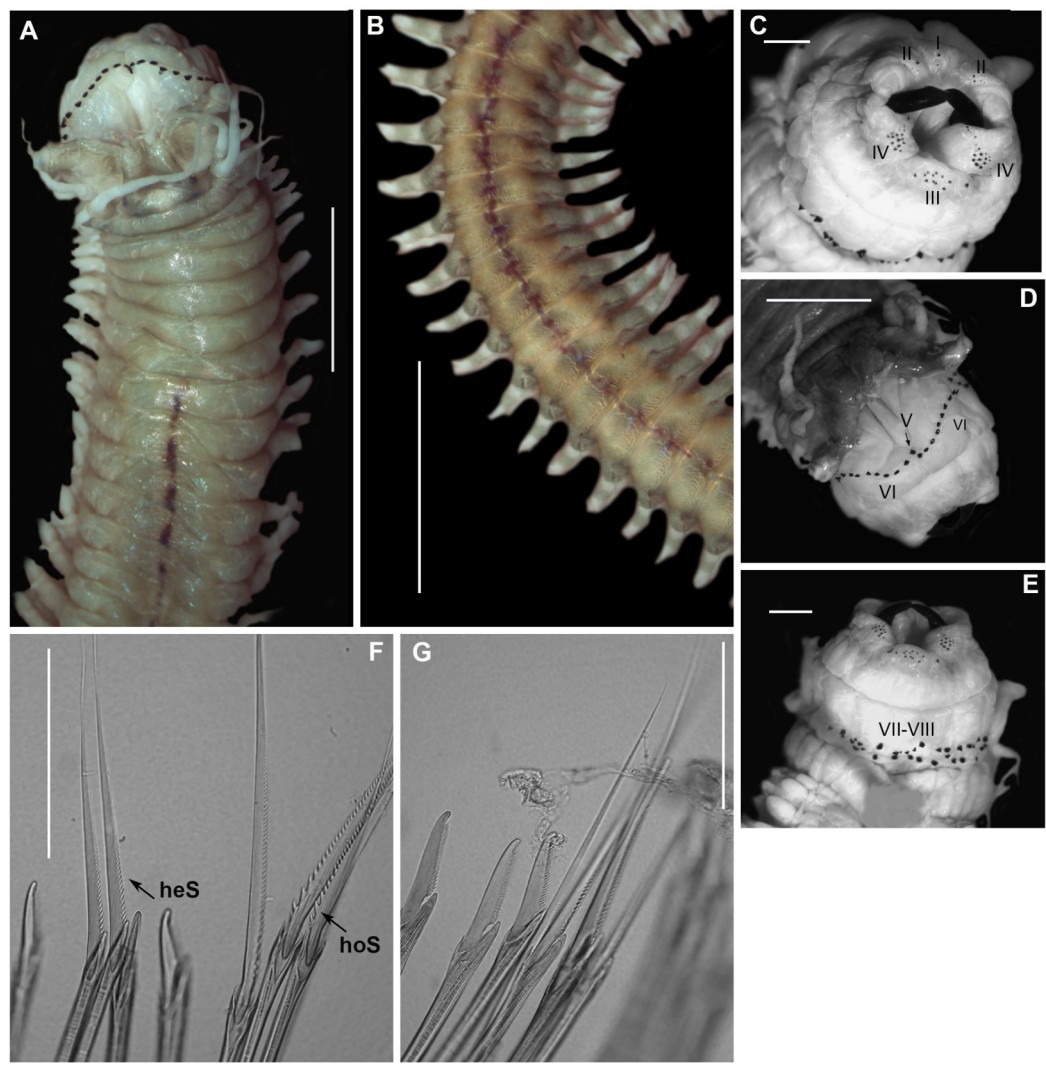

**Figure 20 Morphology of *Platynereis latipalpa* (*Schmarda, 1861*).** (A) Anterior region (dorsal), (B) Middle region showing red colouration on segments (dorsal), (C) Everted pharynx showing Areas 1, 2, 3, 4 (antero-ventral), (D) Everted pharynx showing Areas V and VI (dorsal), (E) Everted pharynx showing Areas VII-VIII (ventro-lateral), (F) Heterogomph spinigers (heS) and homogomph spinigers (hoS) with enlarged teeth at base, (G) Heterogomph falcigers with finely serrated blade. Scale bars: (A) & (B), (D) = 5 mm, (C) & (E) = 2 mm, (F)–(G) = 0.1 mm.

Order: Phyllodocida *Dales, 1962*
Family: Nereididae *Blainville, 1818*
Genus: *Perinereis Kinberg, 1865b*
Species: *Perinereis latipalpa* (*Schmarda, 1861*)
Figure 20

*Nereis (Nereis) latipalpa Schmarda, 1861*: 104–105, txt-fig. A, B, Ka & b, Pl. 31,244
*Neanthes latipalpa Kinberg, 1865b*: 171; *Von Marenzeller, 1888*: 6–7, Fig. 2
*Neanthes latipalpa typica Willey, 1904*: 260–261, Pl. 13, Fig. 9, Pl. 14, Fig. 1–2, 2a & b

*Perinereis nuntia vallata* Day, 1967: 334, Fig. 14.12 P–S; *Branch et al., 2016*: 67, Fig. 25.4 (NOT Grube & Kröyer *in* Grube, 1858).
*Perinereis namibia* Wilson & Glasby, 1993: 265–266, Fig. 10A–K.
*Perinereis latipalpa* Villalobos-Guerrero, 2019: 474–483, Figs. 3–7.
Common name: Coralworm.

*Material examined:* Kommetjie: 34°08′34.5″S 18°19′20.4″E, 3 specimens (complete), MB-A090297–MB-A090299, 10 March 2017, under rocks in the mid-intertidal zone, coll. A. du Toit.

*Description:* Body up to 170 mm. Live specimens, body colour dark green in anterior region, light brown in the middle, to a pale yellow in the posterior. Red blotchy pigment in the middle of each segment, prominent from chaetiger 7–10 onwards (Figs. 20A, 20B, black arrows). Rectangular palpophores with rounded palpostyles. Two antennae, slender with tapering ends (Figs. 20A, 20D). Two pairs of black eyes in a trapezoidal arrangement (Fig. 20A). Maxillary ring with conical paragnaths (Fig. 20C), Area I = 1–2, Area II (variation) = 4(9) + 6(10), Area III = 11–17 in an oval patch, Area IV = 8(33) + 16(32), spoon shaped patch. Oral ring with conical paragnaths (Figs. 20D, 20E), Area V = 1, Area VI = 8(10) + 9(12) in a long arc, Area VII–VIII = 34–58 cones in two irregular rows. Dorsal and ventral cirri present throughout. Notochaetae, homogomph spinigers with serrated blades, first 3 teeth at the base of the blade larger, becoming smaller and uniform till the tip (Fig. 20F, hoS). Neurochaetae, homogomph, heterogomph spingiers with serrated blades, uniform teeth (Fig. 20E, heS) and heterogomph falcigers with medium sized blades, finely serrated (Fig. 20G).

*Remarks:* Specimens collected in this study conformed to the recent redescription in Villalobos-Guerrero (2019). However, variation in body size and paragnath arrangement was noted; total length of paratype is 127 mm and paragnath arrangement, Area III = 9, Area IV = 18-23, Area VII–VIII = 53.

*Collection method:* From under rocks in the mid-intertidal zone.
*Type locality:* Table Bay, Cape of Good Hope, South Africa.
*Known distribution in South Africa:* Hondeklip Bay on the west coast to Port St Johns on the east coast; Namibia: extending north to Luderitz Bay; Mozambique (Day, 1967). However, records in Mozambique have not been confirmed and require further investigation.

Genus: *Pseudonereis* Kinberg, 1865b
Species: *Pseudonereis podocirra* (Schmarda, 1861)
Figure 21

*Mastigonereis podocirra* Schmarda, 1861: 108, Fig. 217.
*Nereis (Nereilepas) stimpsonis* Grube, 1866: 176.
*Pseudonereis variegata* Day, 1967: 331, Fig. 14.12A–F (NOT Grube & Kröyer in Grube, 1858); *Branch et al., 2016*: 66, Fig. 25.1

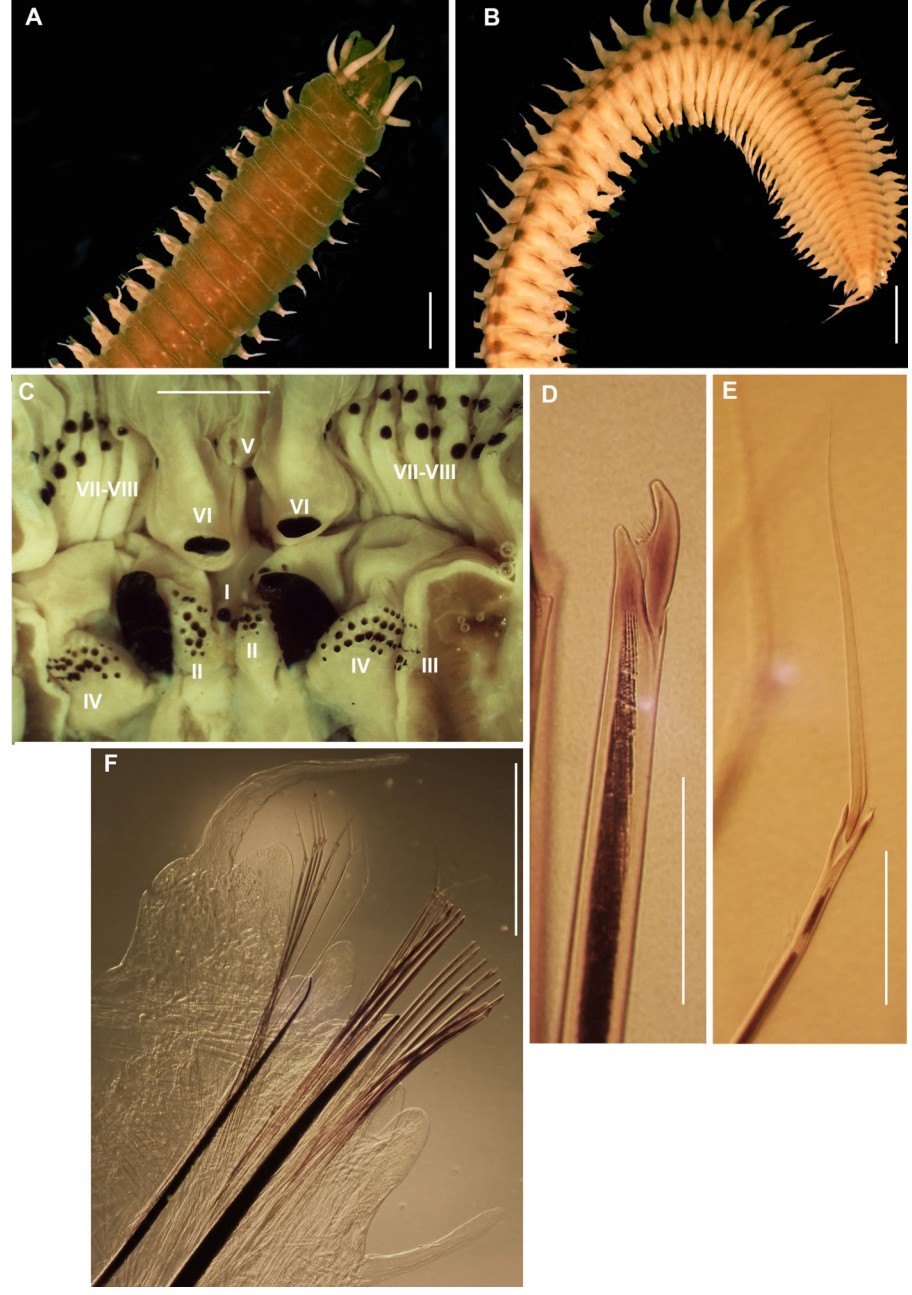

**Figure 21 Morphology of *Pseudonereis podocirra* (*Schmarda, 1861*).** (A) Anterior region (dorsal), (B) Posterior region (dorsal), (C) Paragnaths on pharynx, Areas I–VIII, (D) Compound falciger with serrated blade, (E) Compound spiniger with serrated blade, (F) 30th chaetiger (dorsal). Scale bars: (A)–(C) = 2 mm, (D) & (E) = 0.1 mm, (F) = 1 mm.

*Pseudonereis podocirra Kara, Macdonald & Simon, 2018*: 1286–1291, Figs 2–4
Common name: Musselworm.

*Material examined:* Velddrif: 34°08′34.5″S 18°19′20.4″E, 9 specimens (incomplete), MB-A090355, MB-A090359, MB-A090361–MB-A090363, MB-A090365, MB-A090366,

MB-A090372, MB-A090373, 26 May 2017, from rock pools in the mid-intertidal, coll. A. du Toit. Betty's Bay: 34°22′39.6″S 18°51′21.6″E, 3 specimens (incomplete), MB-A090302, MB-A090304, MB-A090305, 10 February 2017, from under mussel beds in the mid-intertidal mussel belt, coll. A. du Toit. Hermanus: 34°24′41.1″S 19°16′44.8″E, 6 specimens (incomplete), MB-A090306–MB-A090310, MB-A090443, 11 February 2017, from under mussels in the mid-intertidal mussel belt, coll. A. du Toit and H. van Rensburg.

*Description*: Body length up to more than 140 mm. Colour variable: greenish-brown, greyish-brown and medium brown (Figs. 21A, 21B) with white pigmented spots around 4 eyes on prostomium. Black pigmented spots along midpoint of segment boundaries from chaetiger 13 (Fig. 21B). A mix of different types of paragnaths; conical, shield-shaped and p-bars; arranged in distinct areas on pharynx. Area I = 1 conical, Area II = 15–17 conical in a wedge shape, Area III = 22 conical in three or four rows, Area IV = 27–32 conical and p-bars in a closely spaced arc shape, Area V = 1 conical, Area VI = large shield-shaped bars and Area VII–VIII = 40 conical and p-bars alternating in 2–4 rows (Fig. 21C). Oral ring (Fig. 21C), AVI-V-AVI pattern, υ-shaped: ridges of AVI sub-medially separated producing parallel furrows. Notopodial ligule enlarged and elongated from chaetiger 13 to posterior (Fig. 21F). Dorsal and ventral cirri present (Fig. 21F). Homogomph spinigers with finely serrated blades (Fig. 21E) and heterogomph falcigers (Fig. 21D) with concaved and finely serrated blades.

*Remarks*: Specimens collected in the study conformed to the redescription in *Kara, Macdonald & Simon (2018)*, except for body length which was larger, measuring up to a maximum of 140 mm. Molecular analyses (Fig. 22) recovered a single monophyletic group with strong maximum likelihood support, indicating a single genetically similar population, further supporting *Kara, Macdonald & Simon (2018)*. Synonymy of *P. podocirra* with *P. variegata* was recently reversed (*Kara, Macdonald & Simon, 2018*), but it is not known whether *P. variegata* in KwaZulu-Natal in South Africa, Namibia and Mozambique, as reported by *Day (1967)*, are a single species.

*Collection method*: Breaking off mussels by hand from the mussel bed, or by pouring household bleach over the bed (A. du Toit, 2017, personal observation). Collection of nereidid species is no longer permitted (*DAFF, 2017*).
*Type locality*: Cape of Good Hope, Western Cape, South Africa.
*Known distribution in South Africa*: Lamberts Bay to Kidds Beach (*Kara, Macdonald & Simon, 2018*), possibly extending up the east coast to KwaZulu-Natal and Mozambique and up the west coast to Namibia (*Day, 1967*).
*Ecology:* In low intertidal among mussel beds and abandoned *Gunnarea* tubes and barnacle shells.

## DISCUSSION

This study found that more marine annelid taxa are utilised in South Africa as bait than what has previously been reported. In addition to the widely reported and investigated bait species (*Arenicola loveni*, *Gunnarea gaimardi*, *Marphysa haemasoma* and

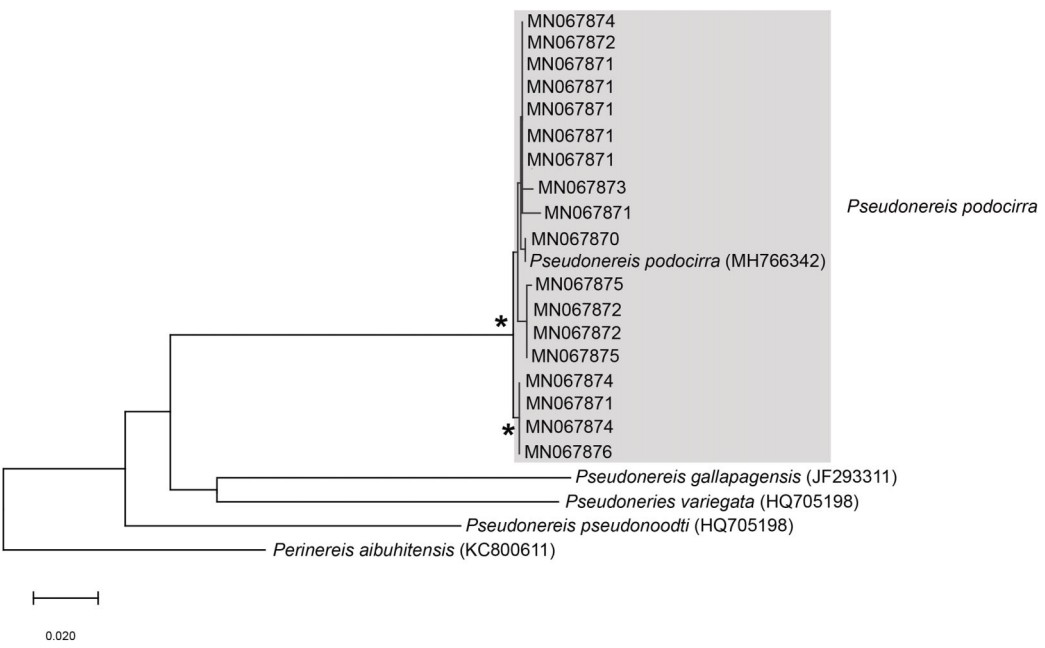

**Figure 22 Neighbour Joining tree of various species belonging to *Pseudonereis Kinberg, 1865b*.**
\* Indicates bootstrap support greater than or equal to 80%. Area highlighted in grey indicates sequences generated in the present study. *Perinereis aibuhitensis* (*Grube, 1878*) was used as an outgroup. Scale bar represents number of substitutions.

*Pseudonereis podocirra*; e.g., *Van Herwerden, 1989*; *Lewis, 2005*; *Sowman, 2006*; *Lewis & Karageorgopoulos, 2008*; *Branch et al., 2016*), several taxa were recorded for the first time (*Abarenicola gilchristi*, *Gunnarea* sp. 1, *Heptaceras quinquedens*, *Lysidice natalensis*, *Marphysa* cf. *corallina*, *Perinereis latipalpa*, *Scoletoma* spp 1 and 2). This is also the first published report of *Siphonosoma dayi* being used, even though there have been anecdotal reports of fishermen collecting sandworm in Knysna since at least 2009 (M.K.S. Smith, 2021, personal communication). By contrast, *Arabella iricolor* and *Eunice aphroditois* (or species matching their general descriptions), which are listed as bait in legislation and field guides (*Marine Living Resources Act, 2014*; *Branch et al., 2016*), were not collected in this study. This suggests that more species are used in the province than collected by us, possibly because these species did not occur at the sites sampled, and or that identifications of these species being used were incorrect. For example, one of the authors never found *E. aphroditois* in the Western Cape Province even after extensive sampling in apparently appropriate substrate, although she did find *Eunice* species in the subtidal in KwaZulu-Natal on the east coast (J. Kara, 2019, 2020, personal observations). Furthermore, *A. iricolor* is superficially similar to lumbrinerid species, and it is possible that both taxa are collected, or that these species were confused in the records for bait collecting. Finally, it is not possible to determine whether species that were collected at single sites (*L. natalensis* and *P. latipalpa*) are targeted more widely, or were misidentified since both were called by names more widely used for other species.
Including *Diopatra aciculata* collected in Knysna in a parallel study (*Van Rensburg, Matthee & Simon, 2020*), 14 species were identified in the Western Cape Province by ten common names, excluding Afrikaans translations. For species collected multiple times and from different locations, individual common names were sometimes applied to more than one species. Species of the same family or genus were often known by a single common name; for example, arenicolids (*Arenicola loveni* and *A. gilchristi*) are bloodworm, onuphids (*D. aciculata* and *H. quinquedens*) are moonshineworms, *Scoletoma* species are puddingworms and *Marphysa* species are wonderworms. For the arenicolids and onuphids this is true even when the species show clear morphological or environmental differences which may have been noted by fishermen, as evidenced by fishermen in Pearly Beach who distinguished between bloodworm (*A. loveni*) and the bakkiewurm (*A. gilchristi*). This was the first time that a second arenicolid is reported as bait, even though *DAFF (2017)* acknowledges that more than one species may be used when they specify that bloodworm are "All species of the genus *Arenicola*", although this is inaccurate as only one species of *Arenicola* has been recorded locally. Individual species were sometimes called by multiple common names that were not translations of the same thing. For example, *M. haemasoma* was identified as wonderworm, bloodworm or bloukoppie (this is Afrikaans for 'blue head', referring to the blue anterior of worms from Knysna and Betty's Bay); *G. gaimardi* was identified as coralworm and polwurm ('pol' is Afrikaans for a tuft, tussock or clump of grass, and may here refer to the clumps of tubes formed by the worms); *P. podocirra* was identified as musselworm and coralworm, while *D. aciculata* was also called the pypiewurm (this is Afrikaans for 'pipe worm', undoubtedly alluding to the chimneys that extend from the mouths of the tubes) by bait collectors in Port Elizabeth (H. van Rensburg, 2017, personal observations). It is also apparent that individual common names were sometimes applied to species from different families, such as coralworm (*G. gaimardi, P. latiplapa¸ P. podocirra*) and musselworm (*P. podocirra, L. natalensis*).

For the most part, subsistence and recreational fishermen used the same names (*e.g.*, for arenicolids, sabellarids, onuphids and *Marphysa* species). Variations in use of names may suggest unfamiliarity with bait worms among some subsistence fishermen, such as bloodworm for *M. haemasoma* in Melkbosstrand, or differences in the use of names depending on geographic region and or type of fishermen, such as coralworm for nereidids at Kommetjie and Velddrif. Interestingly, none of the fishermen used the names from *Branch et al. (2016)* for *M. haemasoma* (estuarine wonderworm which distinguishes it from *E. aphroditois*, the wonderworm), *G. gaimardi* (Cape reef-worm), *L. natalensis* (three-antennaed worm) or *Scoletoma* species (*S. tetraura* false earthworm). Finally, several common names that appear in *DAFF (2017)*, such as rock, shingle, or pot worms, were not used for any of the species collected in this study. The results of this study confirm that common names are sometimes applied in an inconsistent manner by managers and bait collectors. These differences may be maintained through the transfer of knowledge, across generations of bait collectors, of the identification of worms by morphology and ecological patterns. However, it is possible that the application of

common names has changed (*e.g.*, the name moonshineworm applied to onuphids and not *A. iricola* (*Marine Living Resources Act, 2014*)).

The genetic data confirmed the presence of complexes of morphologically similar species within South Africa and globally. *Day (1967)* reported *Gunnarea gaimardi* and *S. tetraura* from Namibia to northern KwaZulu-Natal. Given that this range spans the cold Namaqua, warm Agulhas, and subtropical Natal ecoregions (*Sink et al., 2012*) and barriers to gene flow at Cape Point, Cape Agulhas, Algoa Bay and Wild Coast (*Teske et al., 2011*), it is not surprising that these nominal species each included two genetically distinct species with geographic and habitat separation, respectively. This may also apply to *L. natalensis* that has a similar distribution (*Day, 1967*; *Branch et al., 2016*). Even though all specimens identified here as *Gunnarea* (including *Gunnarea* sp. and *G. gaimardi*) and the *Scoletoma* species from Hermanus and Betty's Bay matched the descriptions of the nominal species provided in *Day (1967*; *G. gaimardi* and *S. tetraura*, respectively*)*, the two genetic groups identified in each could be easily distinguished after thorough morphological examination. This supports *Hutchings & Kupriyanova (2018)* who suggested that many descriptions contained in *Day (1967)*, especially of species described before the 1900s such as the two species under discussion, are too generic to enable accurate identification. Similarly, sequences of *L. natalensis* and *Scoletoma* species 1 and 2 generated in this study do not match those generated for *L. natalensis* and *S. tetraura* collected in India and China, respectively (*Zhou et al., 2010*; *Sigamani et al., 2020*), indicating the presence of complexes of species that may be morphologically similar but genetically distinct, from different locations around the world. *Sigamani et al. (2020)* used *Day (1967)* to identify their samples which also included *H. quinquedens*, originally described from South Africa; unfortunately, we were unable to obtain sequences for the samples that we gathered to test whether the specimens from the two countries are conspecific. However, our results again support *Hutchings & Kupriyanova (2018)* who warned that using *Day (1967)* to identify polychaetes outside of southern Africa may erroneously inflate the distribution ranges of polychaete species.

Resolving the identities of marine annelids used as bait has several important management implications. This is exemplified by the recent discovery that moonshineworm collected in Swartkops and Knysna estuaries is *D. aciculata*, a species originally described in Australia and is probably an alien in South Africa (*Elgetany et al., 2020*; *Van Rensburg, Matthee & Simon, 2020*). Thus, the focus of management of this species must change from conserving populations to preventing further population growth and spread (*Van Rensburg, 2019*; *Van Rensburg, Matthee & Simon, 2020*). This could be done by permitting increased removal by bait collectors, but preliminary investigations suggest that this is unfeasible (*Van Rensburg, 2019*) and that alternative management strategies need to be explored. Knowing the identity of the worms used may also have important implications for the movement of bait species between sites where worms are collected and where fish are caught, since unused bait that can regenerate is frequently discarded in the latter (M.K.S. Smith, South African National Parks, Knysna). This is especially important if the species is alien (as *D. aciculata*), or if species thought to be

locally widespread are multiple species with restricted distributions (as may be the case for *Gunnarea*, *Scoletoma* and *Lysidice* species).

The disjunction between the common names used by collectors and managers is especially problematic when considering the worms that should not be collected. The most recent brochure issued by *DAFF (2017)* states that Cape reef worm (specified as *Gunnarea*), cannot be collected, but that coralworm can. Since collectors contributing to this study all called *Gunnarea* coralworm, and because it is unlikely that many would know the genus name, bait collectors could collect this species not knowing that they are breaking the law (or use it as a defence if they do). The prohibition on collection of *Gunnarea* and musselworms (identified as *Nereis* and *Pseudonereis* by *DAFF (2017)*) is related to the structural damage caused to reefs and mussel beds during collection (*Van Herwerden, 1989*), although this is not clearly articulated in the information brochure. It may therefore be more effective to specify the prohibition of taxa based on the habitats that they occupy, and not just name.

This study was limited by several constraints. Firstly, the geographical coverage was restricted relative to the total coastline of the province; the fishing sites were selected according to where participants could be recruited in advance (because bait collecting is time consuming and needs to coincide with low tides which further limited sampling opportunities, we contacted a core of the participants *via* fishing mailing lists to ensure success in collection) while we also avoided sites that were potentially unsafe, such as Strandfontein and Monwabisi beaches along the northern shores of False Bay. Because of this sampling strategy, there was a bias towards recreational fishermen because subsistence fishermen could not be contacted in advance. Instead, subsistence fishermen were approached on an *ad hoc* basis if they were active at the preselected sampling sites. Additionally, many subsistence fishermen were unwilling to donate bait to the project because bait collecting is so time consuming. We were also reluctant to offer compensation to fishermen because the sale of worms is prohibited by law (*Marine Living Resources Act, 2014*). Consequently, our understanding of the use of common names is still incomplete because species reportedly used as bait, but not found, could not be addressed in this study. This is further exacerbated when fishermen from different fishing sectors and who speak different languages (*e.g.*, English, Afrikaans, isiXhosa) use different names.

In conclusion, the current study has confirmed that more polychaete species are currently used as bait than previously reported. Furthermore, the inconsistent application of common names across taxa and among users, including for the more popular and widespread species, may hamper effective management. The detection of pseudocryptic species complexes among some bait species may have further implications for the management of these taxa as individual species should form separate management units, especially if they are spatially separated. Finally, diversity of marine annelids in general, and bait species in particular, has been underestimated in South Africa, and the global distribution of some has been overestimated. Research to clarify the taxonomy of the members of the pseudocryptic species complexes identified here, *i.e.*, *Scoletoma* species 1 and 2 and *Gunnarea* sp. 1, and the use of polychaetes and common names across a wider geographic range is ongoing.

## ACKNOWLEDGEMENTS

The authors thank Amy Williams for assisting with processing samples, Teresa Darbyshire for commenting on descriptions of the arenicolids, Breyten van der Merwe, Celeste du Toit, the many fishermen who were so generous with their time and expertise, and two anonymous reviewers whose feedback greatly improved the quality of the manuscript.

### Funding

The study was funded by Foundational Biodiversity Information Programme, the Department of Science and Technology and the National Research Foundation awarded to Carol Simon. The funders had no role in study design, data collection and analysis, decision to publish, or preparation of the manuscript.

### Grant Disclosures

The following grant information was disclosed by the authors:
Foundational Biodiversity Information Programme.
Department of Science and Technology and the National Research Foundation: 104890.

### Competing Interests

The authors declare that they have no competing interests.

### Author Contributions

- Carol Simon conceived and designed the sampling strategy, processed samples, prepared figures and/or tables, authored or reviewed drafts of the paper, and approved the final draft.
- Jyothi Kara processed samples, analyzed the data, prepared figures and/or tables, authored or reviewed drafts of the paper, and approved the final draft.
- Alheit du Toit collected and processed samples, prepared figures and/or tables, and approved the final draft.
- Hendré van Rensburg collected and processed samples, prepared figures and/or tables, authored or reviewed drafts of the paper, and approved the final draft.
- Caveshlin Naidoo collected and processed samples, prepared figures and/or tables, and approved the final draft.
- Conrad Matthee assisted with the genetic components of the paper, authored or reviewed drafts of the paper, and approved the final draft.

### Ethics

The following information was supplied relating to ethical approvals (*i.e.*, approving body and any reference numbers):

Stellenbosch University Ethics division approved the involvement of fishermen (SU-HSD-001609).

## Field Study Permissions

The following information was supplied relating to field study approvals (*i.e.*, approving body and any reference numbers):

The Department of Agriculture, Forestry and Fisheries approved the collection of marine invertebrates (RES2017-27).

Stellenbosch University Ethics division approved the involvement of fishermen (SU-HSD-001609).

## Data Availability

The newly generated sequences are available at GenBank: MN045177 to MN045184; MN067870 to MN067882; MN419154 to MN419165; MN419167 to MN419168; MW595992 to MW595995; MW598440 to MW598441. Sequence information can also be found in the Supplemental Information.

The specimens examined are deposited at Iziko Museums of South Africa: MB-A090220 to MB-A090289, MB-A090291 to MB-A090294, MB-A090297 to MB-A090299, MB-A090302, MB-A090304 to MB-A090310, MB-A090310, MB-A090315, MB-A090317, MB-A090318, MB-A090326, MB-A090328, MB-A090331 to MB-A090375, and MB-A090421 to MB-A090443.

## Supplemental Information

Supplemental information for this article can be found online at http://dx.doi.org/10.7717/peerj.11847#supplemental-information.

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
