# Peer review of "Reeling them in: taxonomy of marine annelids used as bait by anglers in the Western Cape Province, South Africa"

_PeerJ, doi:10.7717/peerj.11847_

## Round 0.1 · original submission · Major Revisions

Dear authors,

Two external reviewers have now assessed your manuscript about polychaetes used as bait in South Africa. As you can see, both reviewers have found your study interesting and worthy of publication but have identified some issues that would require careful revision before this manuscript is recommended for acceptance.

Although I have opted for Major Revision, most suggestions were minor but will require some work in the Introduction (e.g. highlighting about the economic value of this bait market in SA), Results and Discussion (e.g. providing separate descriptions for <Gunnarea gaymardi> and <G.> sp. 1) and also improving the quality or re-organizing some of the figures.

I am looking forward to receiving your revised manuscript.

Reviewer 1 ·

Basic reporting

This ms is clear, with a professional English used, even if the text is not always literary. This paper contains a lot of figures but they are all necesseray to illustrate the different results. Most of the figures are excellent but some of them should be improved (see specific comments in the document).

Experimental design

The experimental design is correctly explained, and the difficulty to sample and collaborate with fishermen highlighted. The different questions are well definied and relevant, provided answers about knowledge gap.

Methods are also well explained and rigorous.

Validity of the findings

The results are very interesting for the scientific community and this study should encourage scientists from other countries to check the species used as bait in their own waters.
Results are robust, illustrated both by molecular and morphological data. Some of the descriptions are relatively light but it's not really a problem as no new species is described. Please check my comments directly on the document.

Additional comments

The manuscript by Simon and collabs is a very important contribution to the knwoledge of South African marine worms, highlighted the presence of important diversity and presence of cryptic species among bait worms collected by fishermen.

In the introduction, I would like to have informations about the baits economy in South Africa: are there profesionnal bait collectors? exportation outside SA? Immportation of baits from other countries? Is it only a recreational activity? presence of bait shops?

I think that the discussion should be extended. In the current version, this part is not enough detailed, especially about the next step of these findings. Are you going to describe new species identified with molecular results? How management of these baits could be improved? Importance of correctly identifying the baits (life traits, biodiversity, habitats). Finally, use of live baits could widespread non-indigeneous species, please say some words abouth this situation in South Africa.

Annotated reviews are not available for download in order to protect the identity of reviewers who chose to remain anonymous.

Reviewer 2 ·

Basic reporting

This is a very interesting work on the marine annelids used as bait in the Western Cape, South Africa. It is comprehensive in the sense that the authors are using morphological and molecular evidence to identify the species, in some cases, they even noticed that one single species is hiding at least two species. The finding that common names can represent more than one species is also relevant and it would have surely management implications in South Africa in the near future.

The manuscript is clear and references are sufficient for background and context, although I found many issues in the format and in some cases the rules of the ICZN have not been followed accordingly.

In general, I identified some main issues that the authors should improve. Therefore, I would like to strongly recommend to the authors the following:
1) To perform the description of Gunnarea gaimardi and Gunnarea sp. 1 separately, each with their own figures, instead of giving only one based upon G. gaimardi. The same goes for the two morphospecies found for Scoletoma cf. tetraura.
2) To compare in detail and provide more distinctions between Gunnarea gaimardi and Gunnarea sp. 1, since more morphological differences can be noticed by comparing the figures of both species.
3) To reconsider using the name Marphysa cf. corallina instead of M. corallina (Kinberg, 1865), to compare the M. cf. corallina specimens with descriptions of M. capensis also from South Africa, and to show the maxillary apparatus in figure(s)
4) To evaluate the possible usage of Lysidice atra Schmarda, 1861 over L. natalensis Kinberg, 1865 since they are apparently synonyms and the former claims priority
5) To improve the description of the species, legends of figures, and some figures themselves as suggested in the text.

Other things that the authors must meet is the standardization of the manuscript according to the format of PeerJ or that one they selected where necessary. For instance, lists of synonyms and figures order. All figures must be cited in order in the main text, therefore the figures in the plates must be rearranged.

Experimental design

All the experimental design is clear and well defined. No further comments needed.

Validity of the findings

The impact of the manuscript and novelty is well addressed by the authors. I think a few more details in the discussion on the management implications of their findings will be more meaningful in conservation terms

Additional comments

The authors addressed this relevant topic in a great manner. Systematics of the species using molecular and morphological data is greatly appreciated, although some corrections should be made. Please check my comments on the pdf attached. Best wishes

Annotated reviews are not available for download in order to protect the identity of reviewers who chose to remain anonymous.

---

## Round 0.2 · Minor Revisions

Dear Dr. Simon,

I appreciate you providing point-by-point answers to the reviewers questions and for addressing all of their suggestions. I have read the manuscript and have included some minor editorial suggestions (see attached PDF).

I would like you to pay special attention to the use of S. tetraura sp. 1 and S. tetraura sp. 2. This is very uncommon and should be revisited. Because both molecularly distinct species may not have affinities with 'tetraura' and the current name usage goes against the binomial enforced by the Code, I'd refrain the current use of S. tetraura sp. 1 or S. tetraura sp. 2. One suggestion would be to simply call both taxa Scolotoma sp. 1 and Scolotoma sp. 2 and discuss its similarities to S. tetraura in the Remarks. Another suggestion would be to use Scolotoma sp. 1 (cf. tetraura) and Scolotoma sp. 2 (cf. tetraura) but also make sure you are using the correct syntax (cf. or aff.).

Looking forward to receiving the revised manuscript.

Best,
Wagner Magalhães

---

## Round 0.3 · accepted · Accept

Dear authors,

Many thanks for taking in all of the reviewer's suggestions and also for accommodating my edits.